# PROFILER: BLACK-BOX AI-GENERATED TEXT ORIGIN DETECTION VIA CONTEXT-AWARE INFERENCE PATTERN ANALYSIS

## ABSTRACT

With the increasing capabilities of Large Language Models (LLMs), the proliferation of AI-generated texts has become a serious concern. Given the diverse range of organizations providing LLMs, it is crucial for governments and third-party entities to identify the origin LLM of a given text to enable accurate infringement and mitigation of potential misuse. However, existing detection methods, primarily designed to distinguish between human-generated and LLM-generated texts, often fail to accurately identify the origin LLM due to the high similarity of AI-generated texts from different sources. In this paper, we propose a novel black-box AI-generated text origin detection method, dubbed PROFILER, which accurately predicts the origin of an input text by extracting distinct context inference patterns through calculating and analyzing novel context losses between the surrogate model's output logits and the adjacent input context. Extensive experimental results show that PROFILER outperforms 10 state-of-the-art baselines, achieving more than a 25% increase in AUC score on average across both natural language and code datasets when evaluated against five of the latest commercial LLMs under both in-distribution and out-of-distribution settings.

## 1 INTRODUCTION

As Large Language Models (LLMs) achieve superior capabilities in understanding and generating human-like text, they have become deeply integrated into everyday life (Lo, 2023). However, this growing reliance on LLMs has also raised significant concerns regarding the misuse of AI-generated content (Cotton et al., 2024; Kreps et al., 2022; Perkins, 2023). The European Union's draft Artificial Intelligence (AI) Act (Madiega, 2021) highlights the risks posed by such AI systems, identifying various "high-risk" scenarios where AI misuse could harm fundamental human rights, such as generating phishing emails (Roy et al., 2024). In response, the Act mandates that providers of general-purpose AI models, including LLMs, as well as third-party researchers, develop and implement policies to ensure compliance with copyright laws, so that when severe violations occur, accountability and remediation measures can be enforced effectively.

One key aspect of adhering to these emerging legal and ethical frameworks is the ability to detect the origin of AI-generated text. A large number of detection techniques have recently been developed. Some of these techniques are based on watermarking (Kirchenbauer et al., 2023; Kuditipudi et al., 2024; Hou et al., 2024; Yang et al., 2023). These techniques typically involve fine-tuning LLMs or adjusting their decoding processes to produce text with a distinctive, model-specific distribution. For example, after watermarking, text produced by ChatGPT would exhibit a different distribution from text generated by other LLMs. While watermarking can be effective, it is exclusively controlled by model providers, creating a potential conflict of interest. Since providers are the only entities capable of verifying watermarks, they may be incentivized to obscure evidence of misuse and avoid admitting fault, undermining transparency and accountability.

To mitigate this limitation, surrogate-model-based methods have gained increasing attention (Bhattacharjee & Liu, 2024; He et al., 2023; Wang et al., 2023b). These techniques enable external parties to conduct forensic analyses without requiring cooperation from model providers, relying only on black-box access to the text generation process. By feeding partial or full text to a surrogate model

(i.e., an LLM of a relatively small scale), researchers can analyze its internal states to infer the likely origin of the text. The underlying rationale is that sufficiently powerful surrogate models can capture statistical or representational differences, which help reveal the source. Existing approaches along this line largely focus on identifying next-token prediction patterns, referred to as the *token-level inference pattern*. While these techniques have shown promising results in distinguishing human-generated from AI-generated text, they are less effective in differentiating outputs from various LLMs, as demonstrated in our evaluation (Section 5). Further investigation reveals that, unlike the clear distinction between human and AI-generated text (Jawahar et al., 2020; Bakhtin et al., 2019; Guo et al., 2023), different LLMs often converge on similar next-token predictions due to shared linguistic distributions from large corpora. This similarity introduces a more subtle variation, making token-level inference patterns alone insufficient to capture these nuances (as discussed in Section 3).

Building on this observation, we introduce a novel approach that incorporates contextual information to enlarge the representational differences between text generated by various LLMs, improving the precision of text origin detection. Specifically, rather than relying solely on token-level features (*e.g.*, next token prediction commonly used in existing detection methods), our method broadens the analysis to capture the model's inference behavior over a window of surrounding tokens (*i.e.*, context), referred to as the *context-level inference pattern*. This approach calculates novel context losses by utilizing the output logits from the surrogate model and the adjacent input context tokens at each output logits position. It then extracts both independent features (features derived from a single context loss subsequence) and correlated features (features derived from each pair of context loss subsequences) from these context losses. Based on this, we develop PROFILER, the first black-box detection method that leverages rich contextual information for identifying the origin of AI-generated text. To further evaluate the effectiveness of PROFILER, we extend existing datasets by incorporating diverse text samples generated by multiple recent commercial LLMs across various text domains. Our comprehensive evaluation demonstrates PROFILER's superior performance in detecting text origin.

Our contributions are summarized as follows:

- We propose a novel AI-generated text origin detection algorithm PROFILER that incorporates rich contextual information for improved accuracy.
- We introduce a new feature extraction algorithm that effectively captures contextual information for text origin detection. This algorithm extracts both independent features, i.e., output logits for each token, and correlated features, i.e., pairwise cross-entropy losses between tokens and their neighbors.
- We present a new evaluation dataset for text origin detection, which extends existing datasets by incorporating diverse text samples generated by recent commercial LLMs across various domains and tasks. The dataset includes samples originating from four existing natural language datasets, an existing code dataset, and a newly collected C++ code dataset (GCJ).
- We develop a prototype, PROFILER, and conduct a comprehensive evaluation against 10 baselines. The results demonstrate that PROFILER significantly improves origin detection accuracy, particularly in distinguishing outputs from different LLMs, such as GPT-3.5-Turbo (OpenAI, 2023), GPT-4-Turbo (Achiam et al., 2023), Claude3-Sonnet (Anthropic, 2023), Claude-3-Opus (Anthropic, 2023), and Gemini-1.0-Pro (Team et al., 2023). PROFILER achieves a more than 45.5% and 12.5% increase in detection AUC scores under in-distribution (compared to both zero-shot and supervised-trained methods) and out-of-distribution (compared to supervised-trained methods) settings, respectively.

## 2 BACKGROUND AND RELATED WORK

### 2.1 AI-GENERATED TEXT DETECTION

Existing AI-generated text detection methods can be broadly categorized into two primary approaches: watermark-based methods and surrogate-model-based methods. Watermark-based methods (Kirchenbauer et al., 2023; Kuditipudi et al., 2024; Hou et al., 2024; Yang et al., 2023) typically modify the decoding strategy during the LLM's generation process to force or encourage the generated tokens to fall within a predefined subset of the model's vocabulary. However, these requirements limit the applicability of watermark-based methods, making them less practical compared to surrogate-model-based methods. In contrast, surrogate-model-based methods operate in a completely black-box setting

without requiring prior modifications to the text generation process, where detectors only have access to a limited amount of AI-generated data.

Existing surrogate-model-based detection methods can be further divided into zero-shot detection and supervised-trained detection:

**Zero-shot Detection.** Zero-shot detection methods assign a confidence score to each text sample and use a predefined threshold to differentiate between human-written and AI-generated texts. For instance, GLTR (Gehrmann et al., 2019) measures the average token rank of the input text based on a surrogate LLM's output logits, where a higher average rank indicates a higher likelihood of the text being AI-generated. LRR (Su et al., 2023), an improved version of GLTR, utilizes both log-rank and log-probability metrics. DetectGPT (Mitchell et al., 2023) detects AI-generated texts by measuring the similarity of the input text with its repeatedly masked and reconstructed versions using a pre-trained LLM, while Fast-DetectGPT (Bao et al., 2024) further optimizes this approach with a rapid text sampling technique. Binoculars (Hans et al., 2024) utilizes the cross-entropy between output logits from two different surrogate LLMs to detect AI-generated texts, achieving more consistent performance across LLMs. Other works (Yang et al., 2024; Mireshghallah et al., 2024; Tulchinskii et al., 2023) explore various advanced metrics for zero-shot detection.

**Supervised-trained Detection.** In contrast, supervised-trained detection methods employ more complex features and train classification models to identify distinct patterns in human-written and AI-generated texts. For example, Solaiman et al. (2019) fine-tunes a RoBERTa (Liu et al., 2019) model to detect texts generated by GPT-2 (Radford et al., 2019). RADAR (Hu et al., 2023) and Outfox (Koike et al., 2024) enhance detection robustness against paraphrasing attacks using adversarial training. Raidar (Mao et al., 2024) compares the differences between the original text and LLM-rewritten text to identify AI-generated content. GhostBuster (Verma et al., 2024) explores feature combinations derived from multiple surrogate LLMs' output logits to optimize detection performance. Recent studies (McGovern et al., 2024) also investigate more advanced features for supervised detection.

## 2.2 Black-box Text Origin Detection

Despite the significant advancements in AI-generated text detection techniques, only a few methods have demonstrated the capability to further identify the origin LLM of a given AI-generated text. For example, TuringBench (Uchendu et al., 2021) evaluates the effectiveness of various methods, including GLTR (Gehrmann et al., 2019), Grover (Zellers et al., 2019), and fine-tuning-based approaches (Devlin et al., 2019; Yang et al., 2019) using over 160k samples. However, these methods struggle to keep up with the rapid evolution of LLMs. Sniffer (Li et al., 2023) attempts to detect text origin by comparing the output logits from multiple surrogate LLMs using metrics such as the percentage of perplexity scores. SeqXGPT (Wang et al., 2023a) further enhances Sniffer by leveraging a specialized detection model based on convolutional and self-attention networks. Nevertheless, the effectiveness of these approaches against more advanced commercial LLMs remains uncertain and requires further validation.

## 3 Exploring the Limitation of Existing Detection Methods

The fundamental assumption of existing AI-generated text detection methods is that AI-generated texts exhibit unique next-token prediction patterns, which can be effectively identified using surrogate LLMs. However, these prediction patterns are strikingly similar across texts generated by different LLMs, limiting the effectiveness of such methods in handling text origin detection. Figure 1 illustrates the scores of one latest detector, Binoculars, on texts from human and four distinct LLMs. The x-axis represents Binoculars scores, while the y-axis shows the frequency of samples. Gray bars indicate the score distribution for human-written texts, whereas colored bars represent texts generated by various LLMs. Although the Binoculars score successfully distinguishes between human and AI-generated texts, it shows limited capability in classifying texts based on their specific AI sources. This observation validates our assumption that next-token prediction patterns are highly consistent among different LLMs.

To address the challenge of uncovering distinguishable patterns in AI-generated texts, we propose PROFILER, which goes beyond next-token prediction in the output logits. Figure 2 illustrates the intuition behind our method by comparing text patterns generated by GPT-4-Turbo and Claude-3-

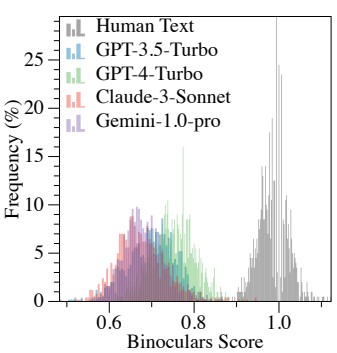

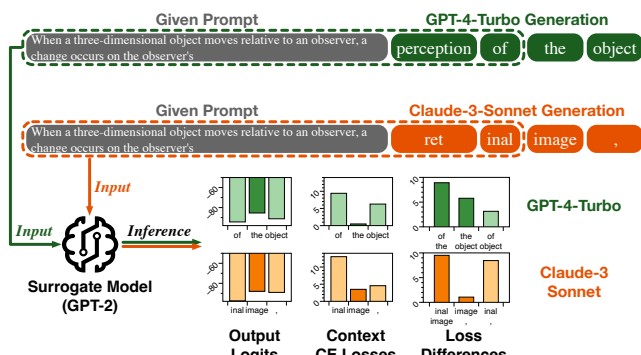

**Figure 1: Distribution difference of Binoculars scores between texts from human and four distinct LLMs.**

**Figure 2: Text patterns generated by GPT-4-Turbo and Claude-3-Sonnet across different metrics. Bars colored in darker colors in the six subfigures highlight features tied to next-token prediction.**

Sonnet. As a standard practice when generating texts using LLMs, a prompt is provided to the model. In this example, both GPT-4-Turbo and Claude-3-Sonnet are given the same prompt, "When a three-dimensional object moves relative to an observer, a change occurs on the observer's". Each model then generates new tokens following its intrinsic pattern, *i.e.*, the texts in green and orange, respectively. During the detection phase, a small surrogate model (*i.e.*, GPT-2 in this example) is used to extract features of the generated texts by inferring them token-by-token, and Profiler analyzes the surrogate model's output logits of those tokens and their cross-entropy losses. The figure shows that given the original prompt (in gray) and part of the generated text (*i.e.*, "perception of" for GPT and "ret inal" for Claude), how Profiler engineers the features. The first feature (*i.e.*, the bar charts in the first column) is the output logits of context. For example, the top-left bar chart shows the output logits of tokens "of", "the", and "object", given the input inside the green dashed box. Ideally, we hope this feature denotes the likelihoods that the model stutters and repeats the previous word "of", correctly predicts the expected word "the", and skips a word and fast-forwards to "object". In contrast, existing techniques only use the logit value of "the". Observe from the two bar charts in the left column that the two features appear similar, meaning that the probabilities follow a similar pattern. To zoom in, Profiler computes the cross-entropy losses between the current output logits (*e.g.*, the logits for "the") and the one-hot encodings of the context (*e.g.*, encodings of "of", "the", and "object", respectively), yielding the charts in the second column. Intuitively, this feature makes the probabilities of stuttering, saying-the-right-word, and skipping more prominent by using the ground-truth tokens as a strong reference. Observe that differences start to emerge. In the last column, we further enhance the distinguishability by subtracting neighboring cross-entropy losses.

## 4 DESIGN OF PROFILER

### 4.1 OVERVIEW OF PROFILER

The entire pipeline of PROFILER consists of three key stages: ***Surrogate Model Inference***, ***Context Loss Computation***, and ***Inference Pattern Extraction***, as shown in Figure 3. The primary objective of PROFILER is to determine whether a given text is generated by a specific text origin.

***Stage 1: Surrogate Model Inference (Section 4.2).*** In this stage, the tokenized input sequence is fed into surrogate model to obtain the sequence of output logits. At each token position, output logits are computed based on all preceding input tokens up to that point.

***Stage 2: Context Loss Computation (Section 4.3).*** With the sequence of output logits from the first stage, PROFILER computes the context loss. At each position, cross-entropy losses between the current output logits and adjacent input tokens within a fixed context window are calculated. These losses, referred to as context losses, are used in the next stage.

**Figure 3: Overview of PROFILER. We take context window size $W = 4$ as an example.**

***Stage 3: Inference Pattern Extraction (Section 4.4).*** Finally, PROFILER extracts inference patterns from the context loss, including independent patterns (statistical and residual patterns of a single context loss) and correlated patterns (distribution similarity between each context loss pair). These patterns are then either used to train a lightweight classifier (e.g., random forest) for text origin detection during the training phase or fed into a pre-trained classifier to obtain predictions.

## 4.2 SURROGATE MODEL INFERENCE

Given the input text to be detected, PROFILER first tokenizes the text and feeds the input tokens into the surrogate model $M$. PROFILER then applies the *Teacher Forcing* algorithm (Williams & Zipser, 1989; Lamb et al., 2016), allowing the surrogate model to infer the input tokens and generate the corresponding output logits sequentially.

Specifically, let the entire input token sequence be $x_{1:n}$, and each component $o_i (i \in \{1, \cdots, n\})$ in the output logits sequence $o_{1:n}$ is calculated as:

$$o_i = P_M(Y_i | X_i = x_{1:i}), \tag{1}$$

where $P_M(Y_i | X_i)$ represents the output logits distribution over $M$'s vocabulary list $V$ at position $i$, given the input token sequence $X_i$.

The output logits sequence $o_{1:n}$ reflects the surrogate model $M$'s next-word or next-few-words predictions, based on its internal knowledge, preferences, and also contains the reduced information of the input tokens up to each position in the sequence. This sequence of output logits $o_{1:n}$ is then used in the next stage to compute the context losses, capturing the inference pattern of the surrogate model with respect to the input text. Notably, though the surrogate model $M$ differ from the origin model of the input text in terms of architecture, size, and training methodology, the potentially overlapping training data, and the powerful statistical and representational understanding capabilities make it a promising tool for uncovering hidden features embedded within the given text.

## 4.3 CONTEXT LOSS COMPUTATION

Compared with existing detection techniques that primarily utilize next-word prediction information contained in the output logits, PROFILER captures and analyzes the information of the surrounding input context at each output position (*i.e.*, inference pattern) by calculating and comparing the cross-entropy losses between each component in the output logits with its adjacent input tokens. These losses are denoted as context losses $\mathcal{L}$. In PROFILER, we use a hyper-parameter $W$ to control the width of the analyzed context at each component of the output logits. PROFILER also drops some of the output logits in $o_{1:n}$ if they lack sufficient context. For example, the first token lacks context from preceding tokens, while the last token lacks context from subsequent tokens. Hence $\mathcal{L} \in \mathbb{R}^{W \times (n-W)}$. Note that, we expect the context to be symmetric (an equal number of preceding and subsequent tokens) in PROFILER, and thus $W$ is always an even number.

For each context loss $\mathcal{L}^j \in \mathcal{L}$ where $j \in \{1, \cdots, W\}$, PROFILER computes its component at each position $i \in \{1, \cdots, n - W\}$ as:

$$\mathcal{L}_i^j = -\sum_{v=1}^{||V||} \tilde{P}_{i-1+j}^v \cdot \log o_{i-1+\frac{W}{2}}^v, \tag{2}$$

where $V$ is the vocabulary of the surrogate model $M$, and $\tilde{P}_k \in \mathbb{R}^{||V|| \times 1}$ is the one-hot encoded vector of input token $x_k$ over the vocabulary list $V$. The calculated context losses $\mathcal{L} = [\mathcal{L}^1, \cdots, \mathcal{L}^W]$ are then used in the next stage to extract the inference pattern.

## 4.4 INFERENCE PATTERN EXTRACTION

With the calculated context losses $\mathcal{L}$, PROFILER then extracts the inference pattern of the surrogate model $M$ regarding the input text $x_{1:n}$, including **independent patterns** and **correlated patterns**.

**Independent Patterns.** For each context loss $\mathcal{L}^j \in \mathcal{L}$, PROFILER first analyzes it independently from other context losses in $\mathcal{L}$. The features extracted from a single context loss are referred to as independent patterns $\mathcal{IP}$, which include both statistical and residual features, representing how each input token in the context is encoded in the output logits during the surrogate model inference. The statistical features $s^j$ of each $\mathcal{L}^j$ consist of six key properties: *average*, *minimum*, *maximum*, *standard deviation*, *median*, and *variance*. The residual features, which are first utilized by PROFILER in AI-generated text origin detection, include the statistical properties of the discrete differences and second-order central differences (Fornberg, 1988; Durran, 2013; Quarteroni et al., 2010) for each context loss $\mathcal{L}^j$. Specifically, the discrete differences $d^j$ for $\mathcal{L}^j$ are calculated as:

$$d_k^j = \mathcal{L}_{k+1}^j - \mathcal{L}_k^j, \text{ for } k \in \{1, \cdots, n - W - 1\}, \tag{3}$$

and the second-order central differences $g^j$ for $\mathcal{L}^j$ are approximated as:

$$g_k^j = \frac{\mathcal{L}_{k+1}^j - \mathcal{L}_{k-1}^j}{2}, \text{ for } k \in \{2, \cdots, n - W - 1\}, \tag{4}$$

where $g_1^j = \mathcal{L}_2^j - \mathcal{L}_1^j$, and $g_{n-W}^j = \mathcal{L}_{n-W}^j - \mathcal{L}_{n-W-1}^j$. Thus, the independent patterns of all the context losses can be represented as $\mathcal{IP} = concat(s^1, \cdots, s^W, \hat{d}^1, \cdots, \hat{d}^W, \hat{g}^1, \cdots, \hat{g}^W)$, where $\hat{d}^j$ and $\hat{g}^j$ represent the statistical properties of the discrete differences $d^j$ and second-order central differences $g^j$, respectively. These components have the same size as the corresponding $s^j$ values.

**Correlated Patterns.** The correlated patterns, denoted as $\mathcal{CP}$, capture how differently the input tokens in the context are encoded in the output logits during surrogate model inference. In PROFILER, we formulate the correlated patterns as the *Symmetric Kullback-Leibler (KL) Divergence* (Moreno et al., 2003) between each context loss pair $\langle \mathcal{L}^j, \mathcal{L}^k \rangle$, which is calculated as:

$$\mathcal{D}_{j,k} = D(\mathcal{L}^{j\prime} || \mathcal{L}^{k\prime}) + D(\mathcal{L}^{k\prime} || \mathcal{L}^{j\prime}), \tag{5}$$

where $\mathcal{L}^{j\prime}$ is the soft-maxed version of $\mathcal{L}^j$ and $D$ represents the KL Divergence (Cover, 1999). Therefore, the correlated patterns $\mathcal{CP}$ consists of $\binom{W}{2}$ *Symmetric KL Divergence* values.

PROFILER finally utilizes the complete inference pattern $[\mathcal{IP}, \mathcal{CP}]$ of the input token sequence $x_{1:n}$ to either train a classifier (*e.g.*, random forest by default in PROFILER) during the training phase or predict the given text's origin during testing.

## 5 EVALUATION RESULTS

### 5.1 EXPERIMENTAL SETTINGS

**Datasets.** To comprehensively evaluate our proposed PROFILER, we use six datasets, consisting of two short natural language datasets, two long natural language datasets, and two code datasets. Specifically, the two short natural language datasets include the Arxiv dataset (Mao et al., 2024) and the Yelp dataset (Mao et al., 2024), which consist of both academic and casual texts. The two long natural language datasets are the Creative dataset (Verma et al., 2024) and the Essay dataset (Verma et al., 2024), which include creative writing samples and student essays, representing fields where LLM misuse is a significant concern. The two code datasets are the HumanEval dataset (Mao et al., 2024; Chen et al., 2021) and the Google Code Jam (GCJ)(Google, 2008-2020; Petrik & Chuda, 2021) dataset, covering short Python code and long C++ code, respectively. Notably, we are the first to introduce a more realistic long C++ code dataset to the AI-generated text detection field. All AI-generated texts are sourced from five of the latest commercial LLMs, including GPT-3.5-Turbo(OpenAI, 2023), GPT-4-Turbo (Achiam et al., 2023), Claude-3-Sonnet (Anthropic, 2023),

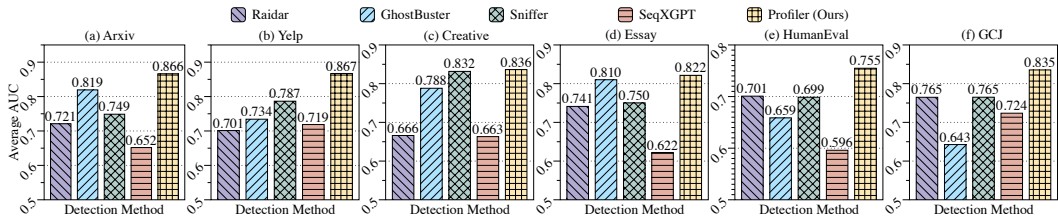

Figure 4: Detection Performance of PROFILER and four supervised-trained baselines on six datasets in out-of-distribution (OOD) setting.

Claude-3-Opus (Anthropic, 2023), and Gemini-1.0-Pro (Team et al., 2023). We also collect the corresponding paraphrased versions of the six datasets following existing studies (Hu et al., 2023) to test the robustness of the detection. More details about the datasets are presented in Appendix B.

**Baselines.** We compare PROFILER with 10 state-of-the-art baselines, including six zero-shot detection baselines and four supervised-trained detection baselines. The zero-shot baselines are LogRank (Gehrmann et al., 2019), LRR (Su et al., 2023), DetectGPT (Mitchell et al., 2023), RADAR (Hu et al., 2023), OpenAI Detector (Solaiman et al., 2019), and Binoculars (Hans et al., 2024). For RADAR and OpenAI Detector, we use their officially released detection models, treating them as zero-shot detectors, even though they were originally designed as supervised-trained detectors. The four supervised-trained detection baselines are Raidar (Mao et al., 2024), GhostBuster (Verma et al., 2024), Sniffer (Li et al., 2023), and SeqXGPT (Wang et al., 2023a), with Sniffer and SeqXGPT officially claiming and evaluating their text origin detection capabilities. We evaluate PROFILER and all baselines in a one-vs-all setting for each text origin, which is a standard evaluation approach in the image origin detection domain (Wang et al., 2024; 2023c) and is suitable for existing baselines since most of them are designed for binary classification tasks.

**PROFILER's Hyper-parameter Settings.** We typically set the context window size $W$ for PROFILER to 6 in most of the experiments, except for the ablation studies. In PROFILER, we employ six open-source LLMs as surrogate models and explore the contribution of each: Llama2-7B (Touvron et al., 2023), Llama2-13B (Touvron et al., 2023), Llama3-8B (Dubey et al., 2024), Mistral-7B (Jiang et al., 2023), Gemma-2B (Team et al., 2024), and Gemma-7B (Team et al., 2024). Notably, these surrogate models are also used by other baseline methods for comparative analysis.

## 5.2 DETECTION PERFORMANCE ON NATURAL LANGUAGE DATASETS

We first evaluate PROFILER against 10 baselines on natural language datasets, including both the original and paraphrased versions of the texts in both the in-distribution and out-of-distribution (OOD) settings. Specifically, under the in-distribution setting, the training and test data are sourced from the same distribution (e.g., both are non-paraphrased samples generated by GPT-3.5-Turbo). In contrast, under the OOD setting, the detectors are trained on the non-paraphrased data but tested on the paraphrased data, providing a more realistic evaluation scenario. The results are presented in Table 1 and Figure 4. Overall, under the in-distribution setting, PROFILER outperforms all 10 baselines, achieving an average improvement of more than 0.30 (45% ↑) in AUC score. Under the OOD setting, PROFILER continues to surpass existing baselines, demonstrating an average AUC score improvement of more than 0.11 (13% ↑). Detailed analysis is shown as follows.

**In-distribution Performance.** The in-distribution performance evaluation results on natural language datasets are presented in Table 1. For each method, we report the 5-fold cross-validated average AUC score. We first evaluate PROFILER alongside 10 baselines on the original dataset. The results highlight the limitations of zero-shot detection methods in identifying the origin of a text, as all zero-shot baselines achieve only around 0.5 average AUC across the six text origins, despite occasionally performing well on specific origins. In contrast, supervised-trained baselines, which leverage more complex features, exhibit significantly better average performance, achieving 0.30 (46% ↑) AUC increase on average. Compared to the zero-shot baselines, PROFILER achieves more than a 0.43 (85% ↑) increase in average AUC score. Additionally, PROFILER outperforms the four supervised-trained baselines by more than 0.10 (12% ↑) in average AUC score. Notably, PROFILER surpasses Sniffer

**Table 1: In-distribution performance comparison on natural language datasets.** Gray color indicates zero-shot baselines, blue and yellow colors indicate supervised-trained baselines, with yellow representing those baselines that officially claim text origin detection capabilities. Our proposed PROFILER is represented by green color.

| | Method | Normal Dataset - In Distribution | | | | | | | Paraphrased- In Distribution | | | | | | |
|---|---|---|---|---|---|---|---|---|---|---|---|---|---|---|---|
| | | Human | GPT-3.5 Turbo | GPT-4 Turbo | Claude Sonnet | Claude Opus | Gemini 1.0-pro | Average AUC | Human | GPT-3.5 Turbo | GPT-4 Turbo | Claude Sonnet | Claude Opus | Gemini 1.0-pro | Average AUC |
| Arxiv | LogRank | 0.8284 | 0.6295 | 0.6515 | 0.4070 | 0.2533 | 0.2320 | 0.5003 | 0.3308 | 0.7447 | 0.6321 | 0.4561 | 0.2287 | 0.6085 | 0.5002 |
| | LRR | 0.1588 | 0.4044 | 0.3611 | 0.5894 | 0.7346 | 0.7501 | 0.4997 | 0.6688 | 0.3161 | 0.3658 | 0.5346 | 0.7099 | 0.4035 | 0.4998 |
| | DetectGPT | 0.8543 | 0.1917 | 0.2544 | 0.5364 | 0.6051 | 0.5566 | 0.4998 | 0.9747 | 0.1706 | 0.2327 | 0.5858 | 0.5999 | 0.4369 | 0.5001 |
| | RADAR | 0.1473 | 0.9229 | 0.4402 | 0.4297 | 0.4561 | 0.6033 | 0.4999 | 0.2030 | 0.8916 | 0.3823 | 0.5168 | 0.3835 | 0.6234 | 0.5001 |
| | OpenAI Detector | 0.3425 | 0.7542 | 0.3277 | 0.4064 | 0.5151 | 0.6537 | 0.5000 | 0.5657 | 0.8234 | 0.3725 | 0.3449 | 0.3714 | 0.5213 | 0.4999 |
| | Binoculars | 0.9789 | 0.4818 | 0.6073 | 0.4596 | 0.2565 | 0.2111 | 0.4992 | 0.7981 | 0.5908 | 0.6100 | 0.3226 | 0.2246 | 0.4544 | 0.5001 |
| | Raidar | 0.8558 | 0.8872 | 0.7739 | 0.6270 | 0.7547 | 0.6801 | 0.7631 | 0.9082 | 0.9024 | 0.7255 | 0.6489 | 0.8095 | 0.7306 | 0.7875 |
| | GhostBuster | 0.9920 | 0.9635 | 0.8878 | 0.7103 | 0.7722 | 0.6873 | 0.8355 | 0.9847 | 0.9765 | 0.8687 | 0.7311 | 0.8255 | 0.6521 | 0.8398 |
| | Sniffer | 0.9875 | 0.9668 | 0.9208 | 0.7296 | 0.8413 | 0.7509 | 0.8662 | 0.9598 | 0.9699 | 0.8733 | 0.7552 | 0.8729 | 0.7331 | 0.8607 |
| | SeqXGPT | 0.9311 | 0.9066 | 0.8763 | 0.6946 | 0.7920 | 0.7343 | 0.8225 | 0.8854 | 0.9054 | 0.8146 | 0.7591 | 0.7903 | 0.6629 | 0.8030 |
| | **PROFILER** | **0.9998** | **0.9809** | **0.9386** | **0.7956** | **0.8815** | **0.8994** | **0.9160** | **0.9998** | **0.9861** | **0.9311** | **0.8870** | **0.9238** | **0.8823** | **0.9350** |
| Yelp | LogRank | 0.6252 | 0.4341 | 0.6154 | 0.3870 | 0.3470 | 0.6025 | 0.5019 | 0.4864 | 0.3792 | 0.6500 | 0.5073 | 0.3871 | 0.6061 | 0.5027 |
| | LRR | 0.4969 | 0.5827 | 0.3893 | 0.5618 | 0.5851 | 0.3692 | 0.4975 | 0.6812 | 0.6580 | 0.3844 | 0.4318 | 0.4879 | 0.3311 | 0.4957 |
| | DetectGPT | 0.3187 | 0.4910 | 0.3958 | 0.5959 | 0.7029 | 0.4952 | 0.4999 | 0.3837 | 0.4096 | 0.2946 | 0.6363 | 0.6817 | 0.6105 | 0.5027 |
| | RADAR | 0.3255 | 0.6400 | 0.3730 | 0.4556 | 0.5660 | 0.6567 | 0.5028 | 0.3979 | 0.7070 | 0.4029 | 0.4248 | 0.5039 | 0.5754 | 0.5020 |
| | OpenAI Detector | 0.3994 | 0.6916 | 0.3021 | 0.4341 | 0.5540 | 0.6329 | 0.5023 | 0.5391 | 0.8226 | 0.3947 | 0.3121 | 0.4235 | 0.5094 | 0.5003 |
| | Binoculars | 0.7820 | 0.4216 | 0.6684 | 0.4042 | 0.2627 | 0.4574 | 0.4994 | 0.6705 | 0.4591 | 0.7041 | 0.3961 | 0.3161 | 0.4464 | 0.4987 |
| | Raidar | 0.9640 | 0.8468 | 0.8108 | 0.7505 | 0.7172 | 0.7578 | 0.8079 | 0.9667 | 0.9117 | 0.7398 | 0.8169 | 0.7287 | 0.7613 | 0.8209 |
| | GhostBuster | 0.8936 | 0.7251 | 0.6829 | 0.6696 | 0.6951 | 0.7509 | 0.7362 | 0.9123 | 0.8245 | 0.7020 | 0.7618 | 0.7547 | 0.6984 | 0.7756 |
| | Sniffer | 0.9236 | 0.7520 | 0.7654 | 0.7127 | 0.7584 | 0.7238 | 0.7726 | 0.9350 | 0.8410 | 0.8059 | 0.7935 | 0.8196 | 0.7544 | 0.8249 |
| | SeqXGPT | 0.8392 | 0.7167 | 0.6940 | 0.6787 | 0.7363 | 0.7110 | 0.7293 | 0.8619 | 0.7873 | 0.7168 | 0.7453 | 0.7609 | 0.7538 | 0.7710 |
| | **PROFILER** | **0.9839** | **0.8563** | **0.8595** | **0.8513** | **0.8758** | **0.8471** | **0.8790** | **0.9881** | **0.9233** | **0.8847** | **0.9071** | **0.8946** | **0.8511** | **0.9081** |
| Creative | LogRank | 0.9201 | 0.1376 | 0.7439 | 0.4138 | 0.2722 | 0.5147 | 0.5004 | 0.7061 | 0.3084 | 0.8241 | 0.4476 | 0.2733 | 0.4244 | 0.4973 |
| | LRR | 0.1450 | 0.8646 | 0.2419 | 0.5560 | 0.7225 | 0.4652 | 0.4992 | 0.4944 | 0.6525 | 0.1475 | 0.5242 | 0.6537 | 0.5350 | 0.5012 |
| | DetectGPT | 0.1949 | 0.6443 | 0.3758 | 0.5760 | 0.6283 | 0.5921 | 0.5019 | 0.3259 | 0.4949 | 0.3564 | 0.6362 | 0.5705 | 0.6489 | 0.5054 |
| | RADAR | 0.0364 | 0.7726 | 0.3109 | 0.5614 | 0.6627 | 0.6797 | 0.5039 | 0.0493 | 0.7105 | 0.3266 | 0.6475 | 0.6022 | 0.7093 | 0.5076 |
| | OpenAI Detector | 0.5389 | 0.7637 | 0.1826 | 0.4189 | 0.5914 | 0.5044 | 0.5000 | 0.7246 | 0.4593 | 0.3379 | 0.4148 | 0.4935 | 0.5880 | 0.5030 |
| | Binoculars | 0.9978 | 0.3854 | 0.7251 | 0.3346 | 0.2542 | 0.2732 | 0.4950 | 0.9722 | 0.3870 | 0.7394 | 0.3519 | 0.2553 | 0.2371 | 0.4905 |
| | Raidar | 0.9209 | 0.8542 | 0.7478 | 0.6888 | 0.6898 | 0.7479 | 0.7749 | 0.8761 | 0.7833 | 0.7796 | 0.7267 | 0.6795 | 0.7233 | 0.7614 |
| | GhostBuster | 0.9847 | 0.9066 | 0.9053 | 0.6865 | 0.7807 | 0.8282 | 0.8487 | 0.9768 | 0.7669 | 0.9079 | 0.7286 | 0.8057 | 0.7592 | 0.8242 |
| | Sniffer | 0.9992 | 0.9256 | 0.9846 | 0.8369 | 0.8527 | **0.9610** | 0.9267 | 0.9979 | 0.9245 | 0.9673 | 0.8225 | **0.8936** | **0.9461** | 0.9253 |
| | SeqXGPT | 0.9682 | 0.8071 | 0.9172 | 0.7397 | 0.7601 | 0.8650 | 0.8429 | 0.9642 | 0.7848 | 0.8788 | 0.7812 | 0.8122 | 0.8510 | 0.8453 |
| | **PROFILER** | **0.9999** | **0.9617** | **0.9935** | **0.9056** | **0.8837** | 0.9307 | **0.9458** | **1.0000** | **0.9558** | **0.9820** | **0.9220** | 0.8898 | 0.9139 | **0.9439** |
| Essay | LogRank | 0.9854 | 0.1349 | 0.7635 | 0.4617 | 0.2719 | 0.3711 | 0.4981 | 0.8642 | 0.3144 | 0.7413 | 0.4175 | 0.1705 | 0.4911 | 0.4998 |
| | LRR | 0.0205 | 0.8804 | 0.2333 | 0.5399 | 0.7377 | 0.5964 | 0.5014 | 0.2467 | 0.6752 | 0.2212 | 0.5850 | 0.7938 | 0.4759 | 0.4996 |
| | DetectGPT | 0.0401 | 0.6341 | 0.4332 | 0.6268 | 0.6306 | 0.6486 | 0.5022 | 0.1165 | 0.5152 | 0.4070 | 0.6778 | 0.6382 | 0.6602 | 0.5025 |
| | RADAR | 0.0151 | 0.8331 | 0.3317 | 0.6718 | 0.6220 | 0.5303 | 0.5007 | 0.0397 | 0.7092 | 0.3176 | 0.7618 | 0.5910 | 0.5891 | 0.5014 |
| | OpenAI Detector | 0.6124 | 0.8426 | 0.1033 | 0.4204 | 0.6024 | 0.4110 | 0.4987 | 0.8874 | 0.5609 | 0.2112 | 0.4392 | 0.5074 | 0.3828 | 0.4982 |
| | Binoculars | 0.9999 | 0.4192 | 0.6470 | 0.2931 | 0.2918 | 0.3348 | 0.4976 | 0.9872 | 0.3682 | 0.7134 | 0.2732 | 0.1603 | 0.4980 | 0.5000 |
| | Raidar | 0.9923 | 0.8843 | 0.8865 | 0.7839 | 0.7646 | 0.7621 | 0.8456 | 0.9698 | 0.8303 | 0.8629 | 0.7670 | 0.7693 | 0.7775 | 0.8295 |
| | GhostBuster | 0.9986 | 0.8992 | 0.8634 | 0.6585 | 0.7648 | 0.8823 | 0.8445 | 0.9927 | 0.7979 | 0.8662 | 0.6655 | 0.8401 | 0.8741 | 0.8394 |
| | Sniffer | 0.9992 | 0.9389 | 0.9938 | 0.8565 | 0.8644 | 0.9398 | 0.9321 | 0.9987 | 0.9306 | **0.9769** | 0.8617 | 0.9190 | 0.9361 | 0.9372 |
| | SeqXGPT | 0.9920 | 0.8258 | 0.9354 | 0.7375 | 0.7273 | 0.8489 | 0.8445 | 0.9674 | 0.8530 | 0.8758 | 0.7616 | 0.8222 | 0.8418 | 0.8536 |
| | **PROFILER** | **1.0000** | **0.9763** | **0.9970** | **0.9297** | **0.9176** | **0.9812** | **0.9670** | **1.0000** | **0.9622** | 0.9748 | **0.9445** | **0.9427** | **0.9728** | **0.9662** |

and SeqXGPT—two supervised-trained baselines specifically designed for text origin detection—by 0.05 (6% ↑) and 0.12 (15% ↑) higher AUC scores on average, respectively.

We further evaluate PROFILER and all baselines on the paraphrased datasets using the same evaluation methodology. Similarly, all zero-shot baselines achieve only around a 0.5 average AUC score, while supervised-trained baselines reach an average AUC of 0.31 (44% ↑). PROFILER outperforms the zero-shot baselines by more than 0.44 (78% ↑) in average AUC and surpasses the supervised-trained baselines by more than 0.11 (12% ↑) on average. While paraphrasing is typically an effective technique to test the robustness of detection methods in the binary AI-generated text detection domain, its impact is reduced in the text origin detection domain, as indicated by the consistent results of supervised-trained baselines and PROFILER across both original and paraphrased datasets. We attribute this to two reasons: 1) all the supervised-trained baselines evaluated in this paper claim to be paraphrasing-robust, and 2) the paraphrasing process might reveal more distinctive characteristics of the specific LLM, providing additional information for text origin detection.

The above results emphasize the superior effectiveness of PROFILER in accurately identifying the origin LLM of a text under the in-distribution setting.

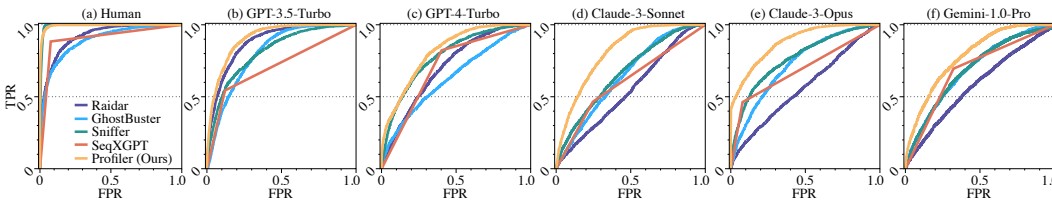

**Figure 5:** ROC curves of PROFILER and four supervised-trained baselines on Yelp dataset in out-of-distribution (OOD) setting.

**Out-of-distribution (OOD) Performance.** Following the in-distribution evaluation, we also assess PROFILER and the baselines under a more realistic out-of-distribution (OOD) setting. Given the poor performance of zero-shot methods in the in-distribution setting, we only consider supervised-trained baselines for the OOD evaluation, shown in Figure 4. We train PROFILER and the four supervised-trained baselines on the original datasets and test them on the paraphrased versions of the same datasets. The OOD experiments aim to evaluate the robustness of the detectors against customized prompts (e.g., paraphrasing prompts in our experiments) used during LLM text generation.

PROFILER outperforms all four baselines across the four natural language datasets, achieving an average AUC improvement of 0.11 (13% ↑). Specifically, under the OOD setting, PROFILER demonstrates a 0.13 (15% ↑) increase in average AUC on the two short natural language datasets (Arxiv and Yelp) compared to the baselines, while exceeding the baselines by more than 0.09 (11% ↑) average AUC on the two long natural language datasets (Creative and Essay). These results not only highlight PROFILER's superior detection performance across different natural language datasets in the OOD setting but also show its significant advantage in handling short text inputs, which are regarded as more challenging in previous studies. More detailed OOD results are presented in Appendix C.

In real-world deployment, detection methods are expected to achieve a high *true positive rate (TPR)* while maintaining a low *false positive rate (FPR)*. Therefore, we further present the ROC curves of PROFILER and the four supervised-trained baselines under the OOD setting using the Yelp dataset in Figure 5. The ROC curves of PROFILER consistently lie above those of the four supervised-trained baselines across different text origins. Specifically, PROFILER achieves an average TPR of over 0.5 when the FPR is less than 0.1. It is important to note that these results are tested under the OOD setting; PROFILER would demonstrate even better performance under the in-distribution setting. Additional ROC curves and detailed analyses for the other five datasets are provided in Appendix F.

## 5.3 Detection Performance on Code Datasets

Existing detection methods are seldom tested on code datasets, despite the growing misuse of LLMs in code generation. We evaluate PROFILER and all baselines on two code datasets: HumanEval (short Python codes) and GCJ (long C++ codes), providing realistic test scenarios. According to the results presented in Table 2 and Figure 4, PROFILER outperforms existing baselines by more than 0.29 (46% ↑) in average AUC score under the in-distribution setting and achieves more than 0.10 (12% ↑) higher average AUC score under the OOD setting on the two code datasets.

**In-distribution Performance.** According to the results presented in Table 2, PROFILER outperforms existing baselines by more than 0.26 (49% ↑) and 0.32 (43% ↑) in AUC scores on the original and paraphrased datasets, respectively, under the in-distribution setting. Specifically, PROFILER surpasses the zero-shot baselines and supervised-trained baselines by 0.34 (68% ↑) and 0.14 (20% ↑) in AUC score on the original dataset, respectively. These results confirm the inadequacy of zero-shot detection scores in the text origin detection domain, as all zero-shot methods only achieve around a 0.5 AUC score on the two code datasets. Furthermore, PROFILER outperforms Sniffer and SeqXGPT with more than 0.16 (25% ↑) and 0.13 (18% ↑) higher AUC scores, respectively, demonstrating its superior effectiveness in detecting the origin of AI-generated code.

The superiority of PROFILER becomes even more evident on the paraphrased dataset, where PROFILER outperforms the zero-shot baselines and supervised-trained baselines by 0.43 (86% ↑) and 0.14 (18% ↑) in AUC score, respectively, across the two paraphrased code datasets. Especially, PROFILER surpasses Sniffer and SeqXGPT with over 0.15 (19% ↑) and 0.14 (18% ↑) AUC scores, respectively.

Table 2: In-distribution performance comparison on code datasets.

| | Method | Normal Dataset - In Distribution | | | | | | | Paraphrased - In Distribution | | | | | | |
|---|---|---|---|---|---|---|---|---|---|---|---|---|---|---|---|
| | | Human | GPT-3.5 Turbo | GPT-4 Turbo | Claude Sonnet | Claude Opus | Gemini 1.0-pro | Average AUC | Human | GPT-3.5 Turbo | GPT-4 Turbo | Claude Sonnet | Claude Opus | Gemini 1.0-pro | Average AUC |
| HumanEval | LogRank | 0.5780 | 0.4297 | 0.4819 | 0.4901 | 0.4401 | 0.5775 | 0.4995 | 0.4049 | 0.4371 | 0.5659 | 0.7525 | 0.4328 | 0.4084 | 0.5003 |
| | LRR | 0.5435 | 0.4932 | 0.4573 | 0.5029 | 0.5448 | 0.4610 | 0.5004 | 0.2467 | 0.6752 | 0.2212 | 0.5850 | 0.7938 | 0.4759 | 0.4996 |
| | DetectGPT | 0.5224 | 0.4311 | 0.4666 | 0.4869 | 0.5347 | 0.5498 | 0.4986 | 0.6482 | 0.4490 | 0.4074 | 0.2810 | 0.6025 | 0.6153 | 0.5006 |
| | RADAR | 0.4960 | 0.5575 | 0.5147 | 0.4955 | 0.4598 | 0.4709 | 0.4991 | 0.3454 | 0.5278 | 0.5474 | 0.7335 | 0.4546 | 0.3932 | 0.5003 |
| | OpenAI Detector | 0.3471 | 0.6882 | 0.5901 | 0.5111 | 0.5118 | 0.3526 | 0.5001 | 0.4913 | 0.8077 | 0.5931 | 0.2185 | 0.4561 | 0.4332 | 0.5000 |
| | Binoculars | 0.6400 | 0.3234 | 0.4175 | 0.5002 | 0.5088 | 0.6134 | 0.5006 | 0.7236 | 0.4116 | 0.5100 | 0.2978 | 0.4221 | 0.6403 | 0.5009 |
| | Raidar | 0.7609 | 0.7944 | 0.6691 | 0.4985 | 0.4619 | 0.6904 | 0.6459 | 0.8438 | 0.8991 | 0.7862 | 0.8563 | 0.7579 | 0.7498 | 0.8155 |
| | GhostBuster | 0.7535 | 0.7547 | 0.6894 | 0.5385 | 0.5510 | 0.7396 | 0.6711 | 0.7078 | 0.8421 | 0.7247 | 0.8527 | 0.6370 | 0.6985 | 0.7438 |
| | Sniffer | 0.7301 | 0.7161 | 0.5404 | 0.4380 | 0.4694 | 0.7117 | 0.6009 | 0.7931 | 0.8039 | 0.6774 | 0.8690 | 0.6641 | 0.7535 | 0.7602 |
| | SeqXGPT | 0.8480 | 0.7542 | 0.6611 | 0.5477 | 0.5296 | 0.7341 | 0.6791 | 0.8534 | 0.8206 | 0.6748 | 0.8496 | 0.5489 | 0.7654 | 0.7521 |
| | **PROFILER** | **0.9366** | **0.8349** | **0.7149** | **0.6720** | **0.7448** | **0.9261** | **0.8049** | **1.0000** | **0.9003** | **0.8602** | **0.9458** | **0.8392** | **0.9166** | **0.9103** |
| GCJ | LogRank | 0.6131 | 0.4212 | 0.5893 | 0.4434 | 0.4395 | 0.4972 | 0.5006 | 0.5161 | 0.3235 | 0.4853 | 0.7185 | 0.5009 | 0.4592 | 0.5006 |
| | LRR | 0.5659 | 0.4230 | 0.3526 | 0.5620 | 0.5328 | 0.5673 | 0.5006 | 0.5002 | 0.7194 | 0.5376 | 0.4552 | 0.2449 | 0.4445 | 0.5995 |
| | DetectGPT | 0.2698 | 0.5817 | 0.7597 | 0.4836 | 0.4951 | 0.4108 | 0.5001 | 0.3808 | 0.6300 | 0.8557 | 0.2609 | 0.4318 | 0.4409 | 0.5000 |
| | RADAR | 0.4600 | 0.6672 | 0.4148 | 0.4724 | 0.4848 | 0.4973 | 0.4994 | 0.3794 | 0.5699 | 0.3801 | 0.6763 | 0.5183 | 0.4778 | 0.5003 |
| | OpenAI Detector | 0.5641 | 0.5268 | 0.4103 | 0.5117 | 0.5207 | 0.4640 | 0.4996 | 0.6472 | 0.6490 | 0.4130 | 0.2611 | 0.4721 | 0.5533 | 0.4993 |
| | Binoculars | 0.7117 | 0.2901 | 0.5866 | 0.4616 | 0.4824 | 0.4721 | 0.5008 | 0.7124 | 0.3899 | 0.7011 | 0.2732 | 0.4400 | 0.4863 | 0.5005 |
| | Raidar | 0.9898 | 0.7704 | 0.7939 | 0.6999 | 0.6638 | 0.8608 | 0.7965 | 0.9852 | 0.8893 | 0.8203 | 0.8864 | 0.7473 | 0.8630 | 0.8653 |
| | GhostBuster | 0.8642 | 0.7652 | 0.6992 | 0.5969 | 0.5872 | 0.7497 | 0.7104 | 0.8638 | 0.8024 | 0.6747 | 0.8314 | 0.6931 | 0.6908 | 0.7594 |
| | Sniffer | 0.9679 | 0.8085 | 0.7362 | 0.6382 | 0.6609 | 0.7524 | 0.7607 | 0.9646 | 0.8260 | 0.7876 | 0.8909 | 0.7230 | 0.7200 | 0.8187 |
| | SeqXGPT | 0.9646 | 0.7990 | 0.6658 | 0.6688 | 0.6657 | 0.7329 | 0.7495 | 0.9529 | 0.8515 | 0.8866 | 0.9212 | 0.7099 | 0.7102 | 0.8387 |
| | **PROFILER** | **0.9966** | **0.9218** | **0.8509** | **0.8119** | **0.7340** | **0.9524** | **0.8780** | **1.0000** | **0.9722** | **0.9804** | **0.9702** | **0.9011** | **0.9616** | **0.9642** |

**Our-of-distribution (OOD) Performance.** Similar to the OOD evaluation on the natural language datasets, we also assess PROFILER and the baselines under the OOD setting on the two code datasets, shown in Figure 4. PROFILER outperforms all four supervised-trained baselines across both code datasets, achieving an average AUC improvement of 0.10 (12% ↑). Specifically, under the OOD setting, PROFILER demonstrates a 0.09 (12% ↑) increase in AUC score on the HumanEval dataset and a 0.11 (11% ↑)increase on the GCJ dataset. More detailed results are provided in Appendix C.

## 5.4 ABLATION STUDY

To investigate the impact of each hyper-parameter on PROFILER 's performance, we conduct several ablation studies, including the effects of context window size and the choice of surrogate model. The results indicate that the hyper-parameters of PROFILER have limited impact on its overall performance, demonstrating the robustness and compatibility of PROFILER across various configurations.

**Impact of Context Window Size.** We evaluate PROFILER using different context window sizes, specifically $W = 2, 4, 6, 8$, where $W = 6$ is the default configuration. The performance of PROFILER fluctuates within a range of 3% across varying window sizes. When $W \leq 6$, a larger window size generally results in a higher average detection AUC. However, when $W \geq 6$, the detection AUC begins to degrade. Therefore, we select $W = 6$ as the default configuration for PROFILER to balance performance and efficiency. More details are presented in Appendix D.

**Impact of Surrogate Model Selection.** We also evaluate the influence of different surrogate models on PROFILER 's performance. While some fluctuation in detection AUC is observed, PROFILER demonstrates consistent performance across various surrogate LLMs. In most cases, using a single surrogate model achieves at least 95% of the detection performance of the ensemble version, indicating PROFILER 's high generality and compatibility when applied with different surrogate models. This flexibility allows PROFILER to be adapted according to different configurations and resource constraints in real-world deployment scenarios. More details are presented in Appendix E.

## 6 CONCLUSION

In this paper, we propose a novel black-box AI-generated text origin detection algorithm that leverages the rich contextual information in the surrogate model's output logits (i.e., inference patterns). Our method comprises three main stages: *surrogate model inference*, *context loss computation*, and *inference pattern extraction*. Extensive experiments on four natural language datasets and two code datasets demonstrate the superiority of PROFILER, achieving more than a 25% average increase in AUC compared to 10 baselines under both in-distribution and out-of-distribution settings.

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

To further demonstrate the effectiveness of our proposed PROFILER and evaluate the contribution of each component in PROFILER, we provide the following supportive materials in the appendix:

- Appendix A presents the visualization of the classification result of PROFILER.
- Appendix B provides additional details about the process of crafting the AI-generated text, including the specific prompts used and the step-by-step procedure. It also presents basic information about the generated datasets, such as the number of samples and the average sample length.
- Appendix C presents the detailed performance comparison of PROFILER and four supervised-trained baselines in OOD setting.
- Appendix D presents detailed ablation study results on the context window size $W$ in PROFILER.
- Appendix E presents the detailed ablation study results on different types of surrogate LLM in PROFILER.
- Appendix F shows the roc curves on other five datasets.

## A    VISUALIZATION OF PROFILER

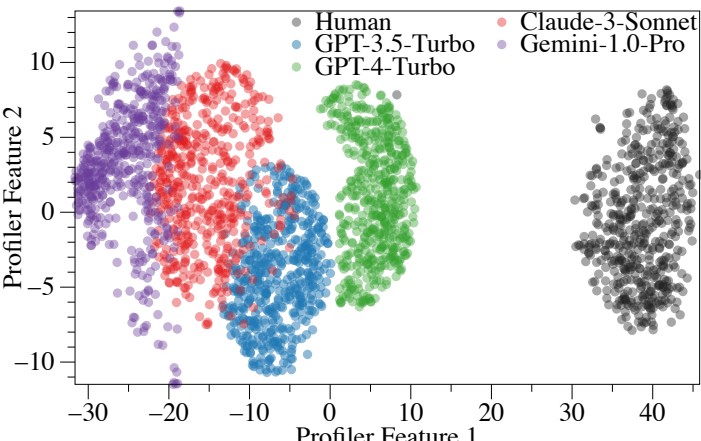

**Figure 6: Visualization of PROFILER's inference patterns on texts from both human and four distinct source LLMs.**

To further validate the effectiveness of our approach, we employ t-SNE (Van der Maaten & Hinton, 2008) to visualize PROFILER 's scores on Essay data in Figure 6. The two axes represent two most representative features extracted by PROFILER. Gray points denote human-written texts, while colored points represent texts generated by different models. Notably, human samples are distinctly separated from AI-generated texts, similar to the performance of Binoculars. However, PROFILER further distinguishes texts generated by different source models, which Binoculars can not. This enhancement is attributed to PROFILER 's incorporation of features that correspond to the contextual tokens in the surrogate model's output logits, remaining distinguishable even among various models.

## B    ADDITIONAL DETAILS OF DATASET CONSTRUCTION

In this paper, we used six datasets in total, including two short natural language datasets (Arxiv and Yelp), two long natural language datasets (Essay and Creative), and two code datasets (HumanEval and GCJ). In this section, we provide further details on how we constructed each of these datasets.

**Arxiv Dataset**. The human-written data is sourced from Mao et al. (2024), which includes 350 abstracts collected from papers published at ICLR between 2015 and 2021. These papers were published before commercial LLMs became publicly available, ensuring that no AI-generated

**Human-written Text** | **AI-generated Text w/o Paraphrasing** | **AI-generated Text w/ Paraphrasing**

**Arxiv**

*We show that deep narrow Boltzmann machines are universal approximators of probability distributions on the activities of their visible units, provided they have sufficiently many hidden layers, each containing the same number of units as the visible layer. We show that, within certain parameter domains, deep Boltzmann machines can be studied as feed forward networks. We provide upper and lower bounds on the sufficient depth and width of universal approximators. These results settle various intuitions regarding undirected networks and, in particular, they show that deep narrow Boltzmann machines are at least as compact universal approximators as narrow sigmoid belief networks and restricted Boltzmann machines, with respect to the currently available bounds for those models.*

*We show that deep narrow Boltzmann machines are universal approximators of probability distributions on the discrete cube. More specifically, we prove that any probability distribution on the discrete cube can be approximated arbitrarily well by a deep narrow Boltzmann machine with a sufficient number of hidden layers. This result holds for any fixed width of the hidden layers, demonstrating the expressive power of depth in these models. Our findings contribute to the understanding of the representational capabilities of deep learning architectures and provide a theoretical foundation for the effectiveness of deep narrow Boltzmann machines in various applications, such as unsupervised learning and generative modeling.*

*In this study, we demonstrate that deep narrow Boltzmann machines possess the remarkable ability to serve as universal approximators of probability distributions on the discrete cube. Specifically, we establish a rigorous proof that any probability distribution on the discrete cube can be approximated to an arbitrary degree of precision by a deep narrow Boltzmann machine, provided that it is equipped with a sufficient number of hidden layers. Notably, this result holds true regardless of the fixed width of the hidden layers, underscoring the immense expressive power that depth confers upon these models. Our groundbreaking findings significantly advance the understanding of the representational capabilities inherent in deep learning architectures and lay a robust theoretical foundation for the impressive effectiveness of deep narrow Boltzmann machines across a wide range of applications, including unsupervised learning and generative modeling.*

**Yelp**

*If you decide to eat here, just be aware it is going to take about 2 hours from beginning to end. We have tried it multiple times, because I want to like it! I have been to it's other locations in NJ and never had a bad experience.*

*The food is good, but it takes a very long time to come out. The waitstaff is very young, but usually pleasant. We have just had too many experiences where we spent way too long waiting. We usually opt for another diner or restaurant on the weekends, in order to be done quicker.*

*Lengthy dining experience with good food*

*This restaurant consistently provides good food but suffers from slow service. Despite multiple attempts to enjoy the dining experience, meals tend to take around two hours from start to finish. While the young waitstaff is generally pleasant, the prolonged wait times have led to diners opting for alternative restaurants, especially on weekends, when a quicker meal is preferred. The restaurant's other locations in NJ seem to offer a better overall experience.*

*This eatery never fails to deliver delectable cuisine, but unfortunately, it is plagued by sluggish service. Despite numerous attempts to savor the dining experience, meals often span a laborious two hours from the first morsel to the final bite. While the youthful waitstaff exudes a pleasant demeanor, the protracted wait times have compelled diners to seek out alternative culinary establishments, particularly on weekends when a more expeditious repast is desired. The restaurant's other Garden State locales appear to provide a more satisfying overall experience.*

**GCJ**

```
/* C Libs */
......

using namespace std;

#define REP(i,s,t) for(int _t=t,i=s;i<_t;i++ )
#define REPP(i,s,t) for(int _t=t,i=s;i<=_t;i++)

template<class T>
void check_max( T&a, T b ){
    if ( a < b ) a = b;
}
template<class T>
void check_min( T&a, T b ){
    if ( a > b ) a = b;
}

//#define debug

map<int,int> mp;
int ans;
void deal( int v ){
    if ( mp[v] <= 1 ) return;
    mp[v-1] ++;
    mp[v+1] ++;
    mp[v]--;mp[v]--;
    ans++;
    deal(v-1);
    deal(v+1);
    deal(v);
}
int main(){
    int T;cin>>T;
    REP(Case,1,T+1){
        int k;cin>>k;
        mp.clear();
        ans = 0;
        REP(i,0,k){
            int v,p;
            cin >>v >>p;
            mp[v] += p;
            deal(v);
        }
        cout << "Case #"<<Case<<": "<<ans<<endl;
    }
    return 0;
}
```

```
#include <iostream>
#include <string>
#include <map>
#include <algorithm>
using namespace std;

#define REP(i, n) for(int i = 0; i < (int)(n); ++i)
#define REPP(i, a, b) for(int i = (int)(a); i < (int)(b); ++i)
template<class T> inline bool check_max(T& a, T b) { if(a
< b) { a = b; return true; } return false; }
template<class T> inline bool check_min(T& a, T b) { if(a
> b) { a = b; return true; } return false; }

map<int, int> mp;
int ans;

void deal(int v) {
    if(mp[v] <= 1) return;
    ++mp[v-1], ++mp[v+1], mp[v] -= 2, ++ans;
    deal(v-1), deal(v+1), deal(v);
}

int main() {
    int T;
    cin >> T;
    REP(t, T) {
        int k;
        cin >> k;
        mp.clear(), ans = 0;
        REP(i, k) {
            int v, p;
            cin >> v >> p;
            mp[v] += p;
        }
        for(auto& pr : mp) deal(pr.first);
        cout << "Case #" << t+1 << ": " << ans << endl;
    }
    return 0;
}
```

```
#include <iostream>
#include <string>
#include <map>
#include <algorithm>
using namespace std;

#define REP(i, n) for(int i = 0; i < (int)(n); ++i)
#define REPP(i, a, b) for(int i = (int)(a); i < (int)(b); ++i)
template<class T> inline bool check_max(T& a, T b) { if(a
< b) { a = b; return true; } return false; }
template<class T> inline bool check_min(T& a, T b) { if(a
> b) { a = b; return true; } return false; }

map<int, int> mp;
int ans;

void handle(int v) {
    if(mp[v] <= 1) return;
    ++mp[v-1], ++mp[v+1], mp[v] -= 2, ++ans;
    handle(v-1), handle(v+1), handle(v);
}

int main() {
    int T;
    cin >> T;
    REP(t, T) {
        int k;
        cin >> k;
        mp.clear(), ans = 0;
        REP(i, k) {
            int v, p;
            cin >> v >> p;
            mp[v] += p;
        }
        for(auto& pr : mp) handle(pr.first);
        cout << "Case #" << t+1 << ": " << ans << endl;
    }
    return 0;
}
```

**Figure 7:** **Examples of human-written text, AI-generated text with paraphrasing, and AI-generated text without paraphrasing from Arxiv, Yelp, and GCJ datasets. We do not show examples from two long natural language datasets here due to the length limit.**

content is mixed into the human-written samples. We utilize the 350 human-written samples to generate AI-generated abstracts using five commercial LLMs: GPT-3.5-Turbo (OpenAI, 2023), GPT-4-Turbo (Achiam et al., 2023), Claude-3-Sonnet (Anthropic, 2023), Claude-3-Opus (Anthropic, 2023), and Gemini-1.0-Pro (Team et al., 2023). Each commercial LLM was given the title of the paper and the first 15 characters of the corresponding human-written abstract, with the prompt used by Mao et al. (2024):

```
The title is {Paper_Title}, start with {Human_Abs[0:15]},
write a short and concise abstract based on this:
```

Each model generated approximately 350 samples, though some models occasionally refused to generate due to their output filtering policies. The average length of both human-written and AI-generated abstracts is approximately 790 characters.

**Yelp Dataset**. For the human-written samples, we use 2,000 Yelp reviews collected from the Yelp Reviews Dataset as compiled by prior work (Mao et al., 2024). To generate the AI-generated data, we utilize five of the latest commercial LLMs, employing the same prompt as used in Mao et al. (2024):

```
Write a concise review based on this: {Human_Review}
```

Each commercial LLM generates ∼2,000 corresponding AI-generated samples, with an average length of fewer than 500 characters. Due to the short average length of the samples, the Yelp dataset is considered the most challenging among all four natural language datasets used in this paper.

**Essay and Creative Datasets**. The human-written samples from both the student essay (Essay) dataset and creative writing (Creative) dataset are both sourced from Verma et al. (2024), each containing 1,000 human samples. To generate the corresponding AI samples, we first use the following prompt to summarize a title from the human-written text:

```
Given the following essay/creative writing, write a title for it:
{Human_Text}
Just output the title:
```

Then, we let the LLM to generate a passage with the summarized title in similar number of words:

```
Write an essay/story in {Length} words to the title:
{Summarized_Title}
```

The procedure and prompts used to generate the Essay and Creative datasets are identical to those used by Verma et al. (2024) and the average character length of the samples in these two datasets exceeds 2,850.

**HumanEval and GCJ Datasets**. The HumanEval and Google Code Jam (GCJ) datasets are two code datasets. HumanEval consists of short Python codes, while GCJ contains long C++ codes. The human-written samples in the HumanEval dataset are sourced from Chen et al. (2021), and the human-written samples in the GCJ dataset are selectively collected from Google (2008-2020). We follow the procedure outlined by Mao et al. (2024) to first generate a description of the purpose and functionality of the human-written codes using the following prompt:

```
Describe what does this code do, including the names and
descriptions of all the functions and global variables:
{Human_Code}
```

Next, we prompt each of the five commercial LLMs to generate corresponding Python or C++ code based on this description:

```
I want to do this:
{Code_Description}
Help me write the corresponding Python/C++ code, no explanation,
just code:
```

Typically, the Python codes generated for the HumanEval dataset are fewer than 50 lines, while the C++ codes generated for the GCJ dataset exceed 100 lines, reflecting their respective complexity and length differences..

**Paraphrased Dataset**. To further test the robustness and transferability of PROFILER and other baselines, we generate six corresponding paraphrased datasets. Following the same procedure as described in Hu et al. (2023), we prompt each commercial LLM to paraphrase its own samples using the following prompt:

```
Enhance the word choices in the sentence to sound more like
that of a human, no explain.
{AI_Sample}
```

We provide concrete examples in Figure 7, including human-written samples, non-paraphrased AI-generated samples, and paraphrased AI-generated samples. Due to the page length limit, we only present samples from two short natural language datasets and one code dataset. Due to space constraints, we include samples from two short natural language datasets and one code dataset. It is evident that distinguishing AI-generated samples from human-written ones without prior knowledge is challenging for humans.

## C   DETAILED PERFORMANCE COMPARISION UNDER OOD SETTING

**Table 3: Detailed performance comparison of PROFILER with four supervised-trained baselines in OOD setting.**

| | Method | Human | GPT-3.5 Turbo | GPT-4 Turbo | Claude Sonnet | Claude Opus | Gemini 1.0-pro | Average AUC | | Human | GPT-3.5 Turbo | GPT-4 Turbo | Claude Sonnet | Claude Opus | Gemini 1.0-pro | Average AUC |
|---|---|---|---|---|---|---|---|---|---|---|---|---|---|---|---|---|
| | | | | | | Paraphrased-OOD | | | | | | | | | | |
| Arxiv | Raidar | 0.7963 | 0.9003 | 0.6739 | 0.5129 | 0.7663 | 0.6764 | 0.7210 | Essay | 0.9474 | **0.7697** | 0.7601 | 0.6444 | 0.6875 | 0.6382 | 0.7412 |
| | GhostBuster | 0.9806 | 0.9772 | **0.8672** | **0.7108** | 0.8113 | 0.5684 | 0.8193 | | 0.9892 | 0.7555 | **0.8741** | 0.5694 | 0.8149 | 0.8573 | 0.8101 |
| | Sniffer | 0.9376 | 0.9510 | 0.6402 | 0.6215 | 0.7851 | 0.5568 | 0.7487 | | 0.9951 | 0.7072 | 0.5399 | 0.6637 | 0.8593 | 0.7361 | 0.7502 |
| | SeqXGPT | 0.8149 | 0.8743 | 0.4900 | 0.5709 | 0.6269 | 0.5340 | 0.6518 | | 0.9774 | 0.5512 | 0.4743 | 0.5556 | 0.5685 | 0.6062 | 0.6222 |
| | PROFILER | **1.0000** | **0.9815** | 0.8009 | 0.6991 | **0.9175** | **0.7991** | **0.8663** | | **1.0000** | 0.6667 | 0.6782 | **0.7796** | **0.8873** | **0.9171** | **0.8215** |
| Yelp | Raidar | 0.9203 | 0.8814 | 0.7073 | 0.5503 | 0.5661 | 0.5818 | 0.7012 | HumanEval | 0.8554 | **0.9054** | 0.7081 | 0.4053 | 0.5676 | **0.7627** | 0.7008 |
| | GhostBuster | 0.8928 | 0.8068 | 0.6361 | 0.6483 | 0.7289 | 0.6923 | 0.7342 | | 0.6614 | 0.8314 | 0.7279 | 0.5175 | 0.5426 | 0.6713 | 0.6587 |
| | Sniffer | 0.9931 | 0.8127 | 0.7841 | 0.6698 | 0.7694 | 0.6898 | 0.7865 | | 0.8319 | 0.8243 | 0.5942 | **0.6179** | 0.6467 | 0.6772 | 0.6987 |
| | SeqXGPT | 0.9041 | 0.7112 | 0.7097 | 0.6095 | 0.6917 | 0.6863 | 0.7187 | | 0.8735 | 0.7549 | 0.6510 | 0.3526 | 0.5191 | 0.4248 | 0.5960 |
| | PROFILER | **0.9947** | **0.9079** | **0.8174** | **0.8140** | **0.8828** | **0.7858** | **0.8671** | | **1.0000** | 0.8927 | **0.8410** | 0.6099 | **0.6766** | 0.5093 | **0.7549** |
| Creative | Raidar | 0.7634 | 0.7128 | 0.7394 | 0.6042 | 0.4629 | 0.7139 | 0.6661 | GCJ | 0.9903 | 0.9266 | 0.7669 | 0.4351 | 0.5847 | 0.8852 | 0.7648 |
| | GhostBuster | 0.9614 | 0.7085 | **0.9043** | 0.6751 | 0.7260 | 0.7533 | 0.7881 | | 0.8606 | 0.7257 | 0.5777 | 0.4222 | 0.5675 | 0.7023 | 0.6427 |
| | Sniffer | 0.9991 | **0.8051** | 0.7589 | 0.7471 | **0.7871** | **0.8930** | 0.8317 | | 0.9934 | 0.9303 | 0.8124 | 0.3798 | 0.6881 | 0.7859 | 0.7650 |
| | SeqXGPT | 0.9400 | 0.5760 | 0.5219 | 0.5793 | 0.5965 | 0.7657 | 0.6632 | | 0.8569 | 0.7974 | 0.8091 | 0.4230 | 0.6763 | 0.7796 | 0.7237 |
| | PROFILER | **1.0000** | 0.7603 | 0.8350 | **0.8294** | 0.7564 | 0.8366 | **0.8363** | | **1.0000** | **0.9548** | **0.8251** | **0.5344** | **0.7939** | **0.9040** | **0.8354** |

Table 3 presents the detailed OOD experimental results of PROFILER on all the six datasets, compared to four supervised-trained baselines. PROFILER outperforms the four baselines in 25 of 36 (70%) cases. Considering the average AUC, PROFILER always reach the best performance, performing 0.8663, 0.8671, 0.8363, 0.8215, 0.7549, and 0.8354 on Arxiv, Yelp, Creative, Essay, HumanEval, and GCJ datasets, individually. Additionally, PROFILER demonstrate great advantage in short natural language datasets (Arxiv and Yelp) and code datasets (HumanEval and GCJ), outperforming the four baselines in 83% and 75% cases, respectively.

Specifically, PROFILER outperforms all four baselines on the Arxiv dataset, achieving an average AUC of 0.8663 and surpassing the next best method (GhostBuster) by 5.74%. Its performance is particularly strong when detecting Human text (AUC = 1.0) and maintaining robustness across various origin LLMs.

On the Yelp dataset, PROFILER demonstrates its effectiveness by achieving the highest average AUC of 0.8671, outperforming the closest baseline, Sniffer, by 9.27%. Its performance remains strong across various LLMs, with perfect detection for Human text and high AUC values for GPT-4-Turbo and Claude-3-Opus.

On the Creative dataset, PROFILER achieves the highest average AUC of 0.8363, marginally outperforming Sniffer by 0.46%. It exhibits consistent performance across diverse LLMs and excels in detecting Human text with a perfect AUC score of 1.0. While Sniffer shows competitive results for a few origin LLMs, its overall lower average AUC and greater variability indicate lower robustness compared to PROFILER.

On the Essay dataset, PROFILER demonstrates its effectiveness by achieving the highest average AUC of 0.8215, marginally outperforming the next best baseline, GhostBuster, by 1.41%. It exhibits stable performance across diverse LLMs, with perfect detection for Human text and high AUC values for Claude-3-Opus and Gemini-1.0-Pro models.

On the HumanEval dataset, PROFILER achieves the highest average AUC of 0.7549, surpassing the next best baseline, Raidar, by 7.72%. It demonstrates robust performance across various origin LLMs and also excels in detecting Human text with a perfect AUC score of 1.0. Although Raidar performs

**Table 4: Ablation study on context window size of PROFILER.**

| | Method | Normal Dataset - In Distribution | | | | | | | Paraphrased- In Distribution | | | | | | |
|---|---|---|---|---|---|---|---|---|---|---|---|---|---|---|---|
| | | Human | GPT-3.5 Turbo | GPT-4 Turbo | Claude Sonnet | Claude Opus | Gemini 1.0-pro | Average AUC | Human | GPT-3.5 Turbo | GPT-4 Turbo | Claude Sonnet | Claude Opus | Gemini 1.0-pro | Average AUC |
| Arxiv | PROFILER W=2 | **0.9998** | 0.9792 | 0.9420 | 0.7938 | 0.8766 | **0.9023** | 0.9156 | 0.9995 | 0.9852 | 0.9360 | 0.8837 | 0.9231 | 0.8759 | 0.9339 |
| | PROFILER W=4 | **0.9998** | 0.9793 | **0.9463** | **0.8012** | 0.8807 | **0.9001** | **0.9179** | 0.9998 | 0.9854 | **0.9400** | 0.8863 | **0.9258** | 0.8790 | **0.9360** |
| | PROFILER W=6 | **0.9998** | **0.9809** | 0.9386 | 0.7956 | 0.8815 | 0.8994 | 0.9160 | 0.9998 | **0.9861** | 0.9311 | **0.8870** | 0.9238 | **0.8823** | 0.9350 |
| | PROFILER W=8 | **0.9998** | 0.9801 | 0.9423 | 0.7970 | **0.8851** | 0.9005 | 0.9175 | **0.9999** | 0.9859 | 0.9334 | 0.8788 | 0.9224 | 0.8772 | 0.9329 |
| Yelp | PROFILER W=2 | 0.9840 | 0.8548 | 0.8563 | 0.8437 | 0.8735 | 0.8509 | 0.8772 | 0.9873 | 0.9135 | 0.8810 | 0.8975 | **0.8946** | 0.8459 | 0.9033 |
| | PROFILER W=4 | 0.9849 | 0.8597 | 0.8619 | 0.8514 | 0.8737 | 0.8507 | 0.8804 | **0.9885** | **0.9240** | **0.8873** | 0.9057 | | **0.8518** | **0.9082** |
| | PROFILER W=6 | 0.9839 | 0.8563 | 0.8595 | 0.8513 | 0.8758 | 0.8471 | 0.8790 | 0.9881 | 0.9233 | 0.8847 | **0.9071** | **0.8946** | 0.8511 | 0.9081 |
| | PROFILER W=8 | **1.0000** | **0.8953** | **0.8817** | **0.9268** | **0.8873** | **0.8574** | **0.9081** | 0.9873 | 0.9222 | 0.8819 | 0.9064 | 0.8923 | 0.8477 | 0.9063 |
| Creative | PROFILER W=2 | **1.0000** | 0.9576 | 0.9924 | 0.8971 | **0.8848** | 0.9255 | 0.9429 | **1.0000** | 0.9501 | 0.9816 | 0.9145 | 0.8914 | 0.9231 | 0.9434 |
| | PROFILER W=4 | **1.0000** | 0.9596 | 0.9932 | **0.9071** | 0.8839 | 0.9298 | 0.9456 | **1.0000** | **0.9572** | **0.9851** | **0.9303** | **0.8956** | **0.9250** | **0.9488** |
| | PROFILER W=6 | 0.9999 | 0.9617 | **0.9935** | 0.9056 | 0.8837 | 0.9307 | **0.9458** | **1.0000** | 0.9558 | 0.9820 | 0.9220 | 0.8898 | 0.9139 | 0.9439 |
| | PROFILER W=8 | 0.9999 | **0.9618** | 0.9929 | 0.9041 | 0.8796 | **0.9325** | 0.9451 | **1.0000** | 0.9557 | 0.9817 | 0.9238 | 0.8881 | 0.9165 | 0.9443 |
| Essay | PROFILER W=2 | **1.0000** | 0.9763 | **0.9975** | 0.9258 | **0.9211** | 0.9821 | 0.9671 | **1.0000** | 0.9609 | 0.9786 | 0.9366 | 0.9438 | 0.9729 | 0.9655 |
| | PROFILER W=4 | **1.0000** | **0.9769** | 0.9970 | **0.9326** | 0.9187 | **0.9823** | **0.9679** | **1.0000** | **0.9655** | **0.9795** | **0.9451** | **0.9452** | **0.9739** | **0.9682** |
| | PROFILER W=6 | **1.0000** | 0.9763 | 0.9970 | 0.9297 | 0.9176 | 0.9812 | 0.9670 | **1.0000** | 0.9622 | 0.9748 | 0.9445 | 0.9427 | 0.9728 | 0.9662 |
| | PROFILER W=8 | **1.0000** | 0.9757 | 0.9967 | 0.9271 | 0.9144 | 0.9813 | 0.9659 | **1.0000** | 0.9612 | 0.9746 | 0.9440 | 0.9406 | 0.9729 | 0.9655 |
| HumanEval | PROFILER W=2 | **0.9497** | 0.8186 | **0.7212** | 0.6827 | **0.7749** | **0.9396** | **0.8145** | 0.9972 | 0.8836 | 0.8377 | 0.9257 | 0.8102 | 0.9130 | 0.8946 |
| | PROFILER W=4 | 0.9423 | 0.8322 | 0.7022 | 0.6694 | 0.7629 | 0.9368 | 0.8076 | **1.0000** | **0.9118** | 0.8472 | 0.9436 | **0.8530** | **0.9219** | **0.9129** |
| | PROFILER W=6 | 0.9366 | **0.8349** | 0.7149 | 0.6720 | 0.7448 | 0.9261 | 0.8049 | **1.0000** | 0.9003 | **0.8602** | 0.9458 | 0.8392 | 0.9166 | 0.9103 |
| | PROFILER W=8 | 0.9378 | 0.8261 | 0.7208 | 0.6562 | 0.7401 | 0.9258 | 0.8011 | **1.0000** | 0.9006 | 0.8568 | **0.9465** | 0.8387 | 0.9101 | 0.9088 |
| GCJ | PROFILER W=2 | **0.9970** | 0.8766 | 0.8173 | 0.7395 | **0.7571** | 0.9068 | 0.8490 | 0.9976 | 0.9624 | 0.9764 | 0.9597 | 0.8870 | 0.8974 | 0.9467 |
| | PROFILER W=4 | 0.9954 | 0.9146 | 0.8480 | 0.7957 | 0.7543 | 0.9317 | 0.8733 | **1.0000** | 0.9729 | **0.9821** | 0.9707 | 0.9014 | 0.9509 | 0.9630 |
| | PROFILER W=6 | 0.9966 | **0.9218** | **0.8509** | **0.8119** | 0.7340 | 0.9524 | 0.8780 | **1.0000** | 0.9722 | 0.9804 | 0.9702 | 0.9011 | 0.9616 | 0.9642 |
| | PROFILER W=8 | 0.9949 | 0.9197 | 0.8501 | 0.8082 | 0.7464 | **0.9584** | **0.8796** | **1.0000** | 0.9735 | 0.9821 | 0.9732 | 0.9018 | **0.9650** | **0.9659** |

well on GPT-4-Turbo, it struggles significantly on Claude 3 models, which underscores PROFILER's superior adaptability and reliability.

On the GCJ dataset, PROFILER achieves the highest average AUC of 0.8354, surpassing the closest baseline, Sniffer, by 9.19%. It demonstrates strong performance across various origin LLMs. Though baselines like GhostBuster perform competitively on some models, their overall lower average AUC and greater variability indicate lower robustness compared to PROFILER.

Overall, the above detailed results under OOD setting confirm PROFILER's superior effectiveness and adaptability in both the natural language origin and code origin detection across various origin LLMs.

# D   DETAILED ABLATION STUDY ON CONTEXT WINDOW SIZE IN PROFILER

Table 4 provides a comprehensive comparison of PROFILER's detection performance across different context window sizes in the OOD setting. The results indicate that the size of the context window significantly influences the system's effectiveness. Generally, employing a larger context window leads to improved AUC scores, especially in datasets like Arxiv and Yelp, underscoring the importance of incorporating more extensive contextual information into the detection process.

However, this trend is not uniform across all datasets. In the HumanEval and Essay datasets, smaller context windows yield comparable or better performance than larger ones. The relationship between context window size and detection performance varies depending on the dataset's characteristics.

These findings highlight the importance of selecting an appropriate context window size tailored to the specific dataset. By adjusting the context window, PROFILER can better capture the most relevant patterns, enhancing its detection capabilities across diverse types of content.

# E   DETAILED ABLATION STUDY ON SURROGATE LLMS IN PROFILER

Table 5 presents a detailed ablation study on the performance of PROFILER across different surrogate LLMs, evaluated on both the normal and paraphrased datasets across various domains. In most cases, the ensemble results outperform those derived from any single surrogate LLM, indicating that combining multiple surrogate models leads to more robust detection performance. However, a significant portion of the detection capability can still be preserved when using individual surrogate models.

**Table 5: Ablation study on surrogate LLMs in PROFILER.**

| | Method | Normal Dataset - In Distribution | | | | | | | Paraphrased- In Distribution | | | | | | |
|---|---|---|---|---|---|---|---|---|---|---|---|---|---|---|---|
| | | Human | GPT-3.5 Turbo | GPT-4 Turbo | Claude Sonnet | Claude Opus | Gemini 1.0-pro | Average AUC | Human | GPT-3.5 Turbo | GPT-4 Turbo | Claude Sonnet | Claude Opus | Gemini 1.0-pro | Average AUC |
| Arxiv | Gemma-2B | 0.9945 | 0.9653 | 0.8670 | 0.7440 | 0.8427 | 0.8764 | 0.8817 | 0.9862 | 0.9714 | 0.8320 | 0.8364 | 0.8677 | 0.8209 | 0.8858 |
| | Gemma-7B | 0.9887 | 0.9715 | 0.8513 | 0.7232 | 0.8366 | 0.8435 | 0.8691 | 0.9706 | 0.9697 | 0.8152 | 0.7788 | 0.8736 | 0.7986 | 0.8678 |
| | Llama2-7B | 0.9995 | 0.9708 | 0.9088 | 0.7777 | 0.8568 | 0.8660 | 0.8966 | 0.9982 | 0.9802 | 0.8919 | 0.8478 | 0.9118 | 0.8334 | 0.9106 |
| | Mistral-7B | 0.9996 | 0.9747 | 0.9060 | 0.7808 | 0.8530 | 0.8734 | 0.8979 | 0.9986 | 0.9782 | 0.8943 | 0.8625 | 0.9052 | 0.8309 | 0.9116 |
| | Llama3-8B | 0.9918 | 0.9704 | 0.9151 | 0.7327 | 0.8532 | 0.8619 | 0.8875 | 0.9633 | 0.9792 | 0.8702 | 0.8199 | 0.8986 | 0.8198 | 0.8918 |
| | Llama2-13B | 0.9977 | 0.9704 | 0.9076 | 0.7709 | 0.8467 | 0.8602 | 0.8923 | 0.9944 | 0.9791 | 0.8974 | 0.8630 | 0.9116 | 0.8234 | 0.9115 |
| | Ensemble | **0.9998** | **0.9809** | **0.9386** | **0.7956** | **0.8815** | **0.8994** | **0.9160** | **0.9998** | **0.9861** | **0.9311** | **0.8870** | **0.9238** | **0.8823** | **0.9350** |
| Yelp | Gemma-2B | 0.8490 | 0.7783 | 0.7546 | 0.7735 | 0.8125 | 0.8057 | 0.7956 | 0.8762 | 0.8254 | 0.7567 | 0.8424 | 0.8126 | 0.7821 | 0.8159 |
| | Gemma-7B | 0.9391 | 0.7799 | 0.7838 | 0.7631 | 0.8150 | 0.7789 | 0.8100 | 0.9345 | 0.8230 | 0.7600 | 0.8219 | 0.8095 | 0.7445 | 0.8155 |
| | Llama2-7B | 0.9664 | 0.8243 | 0.8182 | 0.8246 | 0.8484 | 0.8118 | 0.8489 | 0.9790 | 0.9100 | 0.8553 | 0.8844 | 0.8784 | 0.8175 | 0.8874 |
| | Mistral-7B | 0.9539 | 0.8191 | 0.8095 | 0.8313 | 0.8488 | 0.7885 | 0.8419 | 0.9733 | 0.8869 | 0.8460 | 0.8859 | 0.8734 | 0.8143 | 0.8800 |
| | Llama3-8B | 0.9564 | 0.8168 | 0.8283 | 0.8166 | 0.8595 | 0.7746 | 0.8420 | 0.9555 | 0.8849 | 0.8427 | 0.8713 | 0.8663 | 0.8071 | 0.8713 |
| | Llama2-13B | 0.9744 | 0.8153 | 0.8263 | 0.8269 | 0.8466 | 0.8003 | 0.8483 | 0.9834 | 0.9100 | 0.8583 | 0.8903 | 0.8848 | 0.8142 | 0.8902 |
| | Ensemble | **0.9839** | **0.8563** | **0.8595** | **0.8513** | **0.8758** | **0.8471** | **0.8790** | **0.9881** | **0.9233** | **0.8847** | **0.9071** | **0.8946** | **0.8511** | **0.9081** |
| Creative | Gemma-2B | 0.9935 | 0.8849 | 0.9543 | 0.7950 | 0.7664 | 0.8766 | 0.8785 | **1.0000** | 0.8392 | 0.9472 | 0.8373 | 0.8123 | 0.8725 | 0.8848 |
| | Gemma-7B | 0.9952 | 0.9094 | 0.9536 | 0.7878 | 0.7778 | 0.8403 | 0.8774 | **1.0000** | 0.8589 | 0.9524 | 0.8253 | 0.7948 | 0.8135 | 0.8741 |
| | Llama2-7B | 0.9998 | 0.9308 | 0.9859 | 0.8712 | 0.8354 | 0.8997 | 0.9205 | **1.0000** | 0.8982 | 0.9643 | 0.8652 | 0.8463 | 0.8516 | 0.9043 |
| | Mistral-7B | 0.9999 | 0.9263 | 0.9853 | 0.8683 | 0.8296 | 0.8929 | 0.9170 | **1.0000** | 0.8788 | 0.9625 | 0.8628 | 0.8639 | 0.8711 | 0.9065 |
| | Llama3-8B | 0.9994 | 0.9456 | 0.9809 | 0.8697 | 0.8666 | 0.9140 | 0.9294 | **1.0000** | 0.9372 | 0.9708 | 0.8714 | 0.8715 | 0.8829 | 0.9223 |
| | Llama2-13B | 0.9999 | 0.9316 | 0.9872 | 0.8731 | 0.8450 | 0.9043 | 0.9235 | **1.0000** | 0.9156 | 0.9668 | 0.8605 | 0.8582 | 0.8600 | 0.9102 |
| | Ensemble | **0.9999** | **0.9617** | **0.9935** | **0.9056** | **0.8837** | **0.9307** | **0.9458** | 1.0000 | **0.9558** | **0.9820** | **0.9220** | **0.8898** | **0.9139** | **0.9439** |
| Essay | Gemma-2B | 0.9991 | 0.9119 | 0.9614 | 0.8438 | 0.8258 | 0.9694 | 0.9186 | **1.0000** | 0.8568 | 0.9415 | 0.8797 | 0.8718 | 0.9445 | 0.9157 |
| | Gemma-7B | 0.9991 | 0.8915 | 0.9495 | 0.8146 | 0.8059 | 0.9470 | 0.9013 | **1.0000** | 0.8378 | 0.9204 | 0.8302 | 0.8477 | 0.9417 | 0.8963 |
| | Llama2-7B | **1.0000** | 0.9172 | 0.9913 | 0.8578 | 0.8531 | 0.9307 | 0.9250 | **1.0000** | 0.9166 | 0.9572 | 0.8856 | 0.9140 | 0.9138 | 0.9312 |
| | Mistral-7B | **1.0000** | 0.9513 | 0.9925 | 0.8695 | 0.8643 | 0.9406 | 0.9364 | **1.0000** | 0.8865 | 0.9538 | 0.8863 | 0.9183 | 0.9318 | 0.9294 |
| | Llama3-8B | **1.0000** | 0.9504 | 0.9945 | 0.8805 | 0.8740 | 0.9416 | 0.9402 | **1.0000** | 0.9498 | 0.9668 | 0.9059 | 0.9196 | 0.9341 | 0.9460 |
| | Llama2-13B | **1.0000** | 0.9217 | 0.9922 | 0.8572 | 0.8657 | 0.9356 | 0.9287 | **1.0000** | 0.9380 | 0.9624 | 0.8847 | 0.9212 | 0.9342 | 0.9401 |
| | Ensemble | **1.0000** | **0.9763** | **0.9970** | **0.9297** | **0.9176** | **0.9812** | **0.9670** | 1.0000 | **0.9622** | **0.9748** | **0.9445** | **0.9427** | **0.9728** | **0.9662** |
| HumanEval | Gemma-2B | 0.8614 | 0.7964 | 0.6753 | 0.6406 | 0.6714 | 0.8606 | 0.7510 | **1.0000** | 0.8549 | 0.8231 | 0.9026 | 0.8117 | 0.8841 | 0.8794 |
| | Gemma-7B | 0.8340 | 0.7942 | 0.6720 | 0.6128 | 0.6349 | 0.8559 | 0.7340 | **1.0000** | 0.8660 | 0.8258 | 0.9005 | 0.7900 | 0.8800 | 0.8770 |
| | Llama2-7B | 0.9130 | 0.8186 | 0.7233 | 0.6434 | 0.7227 | 0.9149 | 0.7893 | **1.0000** | 0.8808 | 0.8350 | 0.9330 | 0.7982 | 0.8996 | 0.8911 |
| | Mistral-7B | 0.9297 | 0.8278 | 0.7196 | 0.6338 | 0.7360 | 0.9196 | 0.7944 | **1.0000** | 0.8886 | 0.8487 | 0.9438 | 0.8237 | 0.9020 | 0.9011 |
| | Llama3-8B | 0.7709 | 0.8301 | 0.6745 | 0.4534 | 0.4946 | 0.7786 | 0.6670 | **1.0000** | 0.8958 | 0.8102 | 0.9332 | 0.8121 | 0.8775 | 0.8881 |
| | Llama2-13B | 0.9310 | 0.8192 | 0.7044 | 0.6395 | 0.7335 | 0.9249 | 0.7921 | **1.0000** | 0.8736 | 0.8503 | 0.9375 | 0.8090 | 0.9066 | 0.8962 |
| | Ensemble | **0.9366** | **0.8349** | 0.7149 | **0.6720** | **0.7448** | **0.9261** | **0.8049** | 1.0000 | **0.9003** | **0.8602** | **0.9458** | **0.8392** | **0.9166** | **0.9103** |
| GCJ | Gemma-2B | 0.9734 | 0.8523 | 0.7708 | 0.7609 | 0.6776 | 0.8625 | 0.8163 | **1.0000** | 0.9589 | 0.9785 | 0.9557 | 0.8796 | 0.9124 | 0.9475 |
| | Gemma-7B | 0.9730 | 0.8658 | 0.7626 | 0.7452 | 0.6849 | 0.8642 | 0.8160 | **1.0000** | 0.9476 | 0.9749 | 0.9462 | 0.8743 | 0.8942 | 0.9396 |
| | Llama2-7B | 0.9912 | 0.8913 | 0.8123 | 0.7737 | 0.7010 | 0.8827 | 0.8420 | **1.0000** | 0.9642 | 0.9697 | 0.9498 | 0.8593 | 0.9161 | 0.9432 |
| | Mistral-7B | 0.9959 | 0.8945 | 0.7931 | 0.7760 | 0.7282 | 0.8956 | 0.8472 | **1.0000** | 0.9679 | 0.9733 | 0.9687 | 0.8782 | 0.9128 | 0.9502 |
| | Llama3-8B | 0.9687 | 0.8739 | 0.7890 | 0.7757 | 0.6913 | 0.9354 | 0.8390 | **1.0000** | 0.9608 | 0.9564 | 0.9473 | 0.8863 | 0.9228 | 0.9456 |
| | Llama2-13B | 0.9913 | 0.9023 | 0.8114 | 0.7960 | 0.7134 | 0.8866 | 0.8502 | **1.0000** | 0.9614 | 0.9644 | 0.9434 | 0.8731 | 0.9185 | 0.9435 |
| | Ensemble | **0.9966** | **0.9218** | **0.8509** | **0.8119** | **0.7340** | **0.9524** | **0.8780** | 1.0000 | **0.9722** | **0.9804** | **0.9702** | **0.9011** | **0.9616** | **0.9642** |

Among all the surrogate LLMs, the Llama series models—including Llama2-7B, Llama3-8B, and Llama2-13B generally perform the best across different datasets. For instance, on the *Arxiv* dataset, Llama2-7B achieves an average AUC of 0.8966 on the normal dataset and 0.9106 on the paraphrased dataset, outperforming the Gemma series models. Similarly, on the *Creative* dataset, Llama2-13B attains an average AUC of 0.9235 on the normal dataset and 0.9102 on the paraphrased dataset.

In contrast, the Gemma series models tend to underperform compared to the Llama series. For example, Gemma-2B achieves an average AUC of 0.8817 on the *Arxiv* normal dataset and 0.8858 on the paraphrased dataset, which is lower than the corresponding results from Llama2-7B and Mistral-7B. Notably, the performance differences among surrogate LLMs are not strictly correlated with model size. For example, on the *HumanEval* dataset, Mistral-7B achieves an average AUC of 0.9011, which is higher than both Llama2-7B (0.8911) and the larger Llama2-13B (0.8962). The ensemble approach consistently yields better performance across all datasets than individual surrogate LLMs in most cases. This suggests that leveraging the strengths of multiple models can effectively enhance the detection capabilities of PROFILER.

Overall, PROFILER shows consistent performance across different surrogate LLMs, demonstrating the compatibility of our method and indicating that PROFILER is able to effectively work with various surrogate models without significant loss in performance.

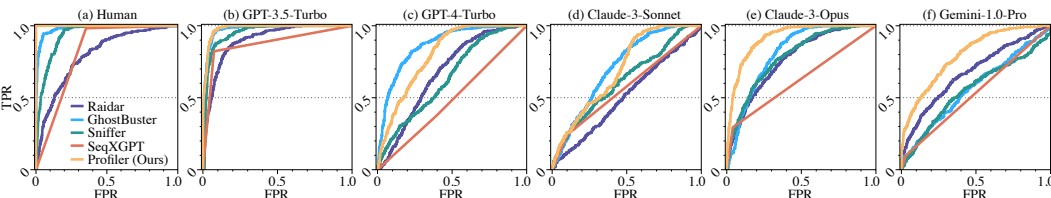

**Figure 8:** ROC curves of PROFILER and four supervised-trained baselines on Arxiv dataset in out-of-distribution (OOD) setting.

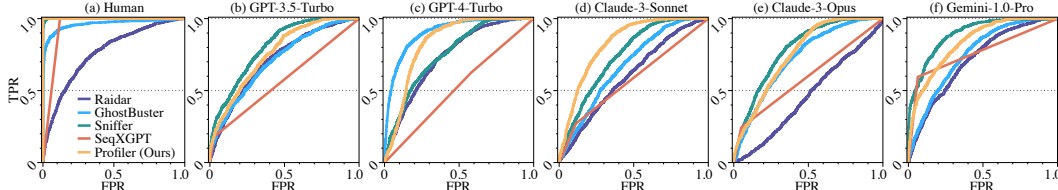

**Figure 9:** ROC curves of PROFILER and four supervised-trained baselines on Creative dataset in out-of-distribution (OOD) setting.

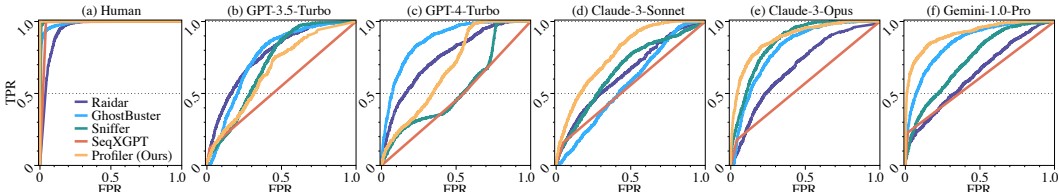

**Figure 10:** ROC curves of PROFILER and four supervised-trained baselines on Essay dataset in out-of-distribution (OOD) setting.

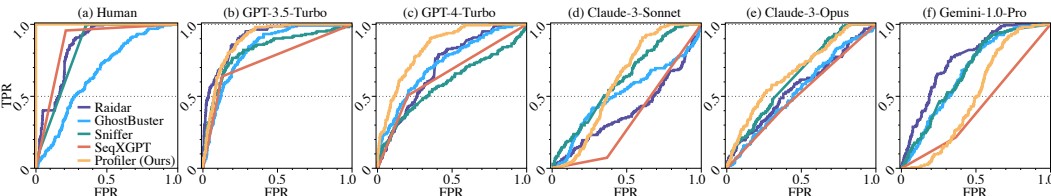

**Figure 11:** ROC curves of PROFILER and four supervised-trained baselines on HumanEval dataset in out-of-distribution (OOD) setting.

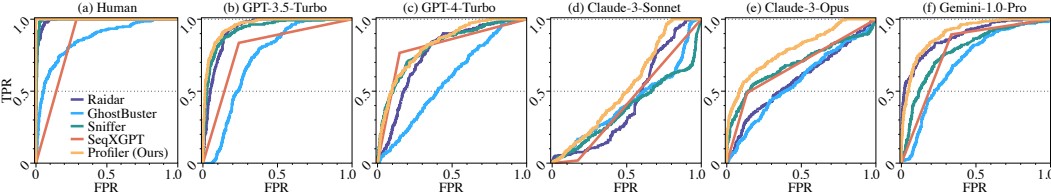

**Figure 12:** ROC curves of PROFILER and four supervised-trained baselines on GCJ dataset in out-of-distribution (OOD) setting.

# F  ADDITIONAL ROC CURVES UNDER OOD SETTING

We present the OOD ROC curves of PROFILER and four supervised-trained baselines in Figure 8 (Arxiv), Figure 9 (Creative), Figure 10 (Essay), Figure 11 (HumanEval), and Figure 12 (GCJ) individually. Similar to its performance on the Yelp dataset, PROFILER ranks first or second in 63% of the cases. Additionally, PROFILER demonstrates a significant advantage when operating in the low false positive rate (FPR) mode, achieving over 0.4 true positive rate (TPR) when the FPR is restricted to just 0.1. It is noteworthy that these ROC curves are calculated under the OOD setting.

The performance gap of PROFILER in the low FPR mode would be even more pronounced under the in-distribution setting, highlighting its effectiveness in distinguishing the text origin with minimal false positives.

