# OpenReview forum: "Profiler: Black-box AI-generated Text Origin Detection via Context-aware Inference Pattern Analysis"
_ICLR.cc/2025/Conference — Submitted to ICLR 2025_

### Official Review · Reviewer_1fGG · 2024-10-29

**Soundness:** 3
**Presentation:** 4
**Contribution:** 3
**Rating:** 6
**Confidence:** 5

**Summary:**

This paper addresses the challenge of detecting the origin of AI-generated texts, given the increasing capabilities of large language models (LLMs) and the similarity of texts produced by different models. Current detection methods struggle to accurately identify the specific source model. To tackle this, the authors propose PROFILER, a novel black-box detection method that predicts the origin of a text by analyzing distinct context inference patterns, specifically by calculating context losses between the surrogate model’s output logits and adjacent input contexts.

**Strengths:**

1. The paper is well-written and presents its ideas clearly, making it accessible to readers from both technical and non-technical backgrounds.
2. The detection method is effective and rigorously tested. The authors designed comprehensive experiments, evaluating the model against ten state-of-the-art baselines and providing performance comparisons under both in-distribution and out-of-distribution scenarios.
3. Unlike prior methods that primarily focus on distinguishing human-generated from AI-generated texts, this work addresses the more nuanced task of identifying the specific source model.

**Weaknesses:**

The paper does not have any major shortcomings, but please refer to the Questions session to add additional analysis of experimental observations.

**Questions:**

1. In Table 1, I was surprised by the significant performance variation of the baseline methods implemented by the authors across different models, **ranging from 0.01 to 0.8**. In contrast, PROFILER appears to perform more robustly. Could the authors provide further analysis of this performance variation and discuss potential reasons why PROFILER appears more robust across models

2. Similarly, in Table 2, I noticed that the scores of the two models from the Claude family are relatively lower compared to other models in the Normal Dataset. Could the authors provide more discussion on this observation? Additionally, it would be helpful to explain why the performance of your method is better on paraphrased datasets than on the Normal Dataset.

---

> ### Author Response · Authors · 2024-11-27
> **Response to Reviewer 1fGG**
>
> Thanks for your insightful review. Here are our point-by-point responses:
>
> > Q1: In Table 1, I was surprised by the significant performance variation of the baseline methods implemented by the authors across different models, ranging from 0.01 to 0.8. In contrast, PROFILER appears to perform more robustly. Could the authors provide further analysis of this performance variation and discuss potential reasons why PROFILER appears more robust across models?
>
> The high performance variation (e.g., from 0.01 to 0.8) observed in Table 1 and Table 2 primarily occurs in zero-shot detection methods. In contrast, supervised-trained detection methods generally demonstrate more robust performance across different source models, with detection AUCs typically exceeding 0.65. The main reason for this difference lies in the nature of the features these methods utilize. Zero-shot detection methods assign a single score to a given text. Such a single score can hardly separate texts from multiple sources, especially for those texts generated by different source LLMs. In this case, texts generated by part of the source LLMs used in our experiments may have similar zero-shot scores, causing the high variation in the detection performance of Zero-shot methods.
>
> Supervised-trained methods, on the other hand, leverage multi-dimensional feature vectors that enable them to extract more complex and subtle differences across texts generated by different LLMs. Thus, all the supervised-trained methods perform more robustly across texts generated by different LLMs, supported by much lower variance than zero-shot methods. While our Profiler performs the best due to the more effective features extracted by it.
>
> ---
>
> > Q2: Similarly, in Table 2, I noticed that the scores of the two models from the Claude family are relatively lower compared to other models in the Normal Dataset. Could the authors provide more discussion on this observation? Additionally, it would be helpful to explain why the performance of your method is better on paraphrased datasets than on the Normal Dataset.
>
> Thank you for highlighting these interesting points. The consistently lower detection AUC scores on the Claude family models are observed not only in baseline detection methods but also in Profiler. This may be because of two reasons: (1) The evaluation setting used in our paper. (2) The similarity between the texts generated by Claude models.
>
> In our paper, we evaluate the detection performance of each detector using one-vs-all setting across texts generated by different LLMs, where we take texts generated by all the sources into comparison but we only label texts generated by one specific source as positive at each time. For example, when we test the origin detection performance of the detectors toward GPT-3.5-Turbo, we include the human-written texts and texts generated by all five LLMs. While we only label the texts generated by GPT-3.5-Turbo as positive, the human-written texts and texts generated by the other four LLMs are labeled negative. Thus, considering Claude-3-Sonnet and Claude-3-Opus are models in the same generation (both Claude-3 generation), their texts may have similar patterns to the detectors, causing performance drops on both models in our evaluation setting.
>
> Regarding the occasionally better performance of Profiler (and also other baselines) on paraphrased datasets in Table 1 and Table 2, it is important to note that these are in-distribution results, where the training and test data distributions are the same. When detectors are tested in an out-of-distribution setting—where the detector is trained on the original dataset and tested on the paraphrased dataset—all detectors exhibit a performance degradation, as shown in Figure 4.
>
> The improved performance on paraphrased datasets under the in-distribution setting suggests that paraphrased data is more separable in this context. We attribute this to two main reasons: (1) paraphrasing may inadvertently expose more model-specific characteristics, and (2) different LLMs may interpret and encode patterns of human-written texts differently, thereby reducing detection complexity.
>
> However, the performance drop observed in the out-of-distribution setting indicates that paraphrasing remains an effective evasion technique in real-world deployments.

---

> ### Author Response · Authors · 2024-12-02
> **Kind Reminder from Authors**
>
> Dear Reviewer 1fGG,
>
> We would like to express our sincere appreciation for your valuable suggestions, which have significantly improved the quality of our manuscript. In response to your feedback, we have made our best effort to address your concerns about more detailed analysis of the experimental observations by providing more detailed explanations in our rebuttal response.
>
> We would be grateful for any further feedback you may have on the revised version and our responses. If there are any aspects that remain unclear, we are more than willing to provide additional clarification.
>
> Thank you once again for your time and thoughtful review. We look forward to your response.
>
> Best regards,
>
> The Authors

---

### Official Review · Reviewer_bf21 · 2024-11-03

**Soundness:** 2
**Presentation:** 2
**Contribution:** 2
**Rating:** 3
**Confidence:** 5

**Summary:**

PROFILER proposes a novel method for detecting the origin of AI-generated text using a black-box approach that involves calculating novel context losses between the output logits of a surrogate model and the adjacent input context. PROFILER can differentiate texts generated by various LLMs with higher precision by broadening the analysis beyond simple next-token prediction patterns to include contextual information around each output token.

**Strengths:**

Overall, the proposed PROFILER has the following advantages:
1. PROFILER consistently outperforms state-of-the-art baselines, showing over 25% average increase in AUC score across evaluations, indicating a robust capability to detect the origin of AI-generated texts.
2. Effective across both natural language and code datasets, demonstrating the method’s adaptability to different content types.
3. The use of context-level inference patterns provides a deeper insight into the generation patterns of different LLMs, improving discrimination between sources.

**Weaknesses:**

1. Some terms, concepts, and figure captions need definitions and explanations for readers to better understand.
2. Mathematical notations and derivation need improvement.
3. The experimental setup requires enhancement, and further validation is necessary to evaluate its generalizability.

**Questions:**

1. In lines 69-70, references are needed for independent and correlated features. 'output logits for each token' needs further explanation. Are they feature vectors learned from something? Why are they independent?
2. In lines 184-186, do the PROFILER features only have 2 dimensions? If not, how are these two dimensions selected?
3. In Figure 2, most of the features have an oval shape. This makes sense since projecting them onto the PROFILER feature axis 1/2 gives you Gaussian distributions. Is there an explanation for why do the GPT-3.5 Turbo features (green) not follow a 2D Gaussian distribution and why does it look very different from GPT-4 Turbo (I do not expect the shapes to be very different from GPTs)?
4. It seems PROFILER feature 1 provides most of the separable information, and the feature 2 ranges of different data samples are highly overlapped. Is it possible to separate the texts with only one feature?
5. In Equation 1, what does the black dot represent? Functions with black dots represent a family of functions and are usually used for caption explanations, but not used for formally defining a variable.
6. Lines 235-240, what is the definition of the input token sequence X? X=x_{1:n} or X=x_{1:i}?
If X=x_{1:i}, X needs a subscript index i (e.g., X_i), since X depends on i.
7. Line 256, what is the definition of C? Is C a set or a vector?
8. Lines 259-260, why does an even W guarantee the context matrix L^C to be symmetric? It seems that the dimension of L^C also depends on n.
9. Equation 2 looks confusing. Considering that tilde P_k is a ||V||x1 vector, the dot in Equation 2 is an element-wise product (if o^v_{i-1+W/2} is a scalar) or a vector inner product (if o^v_{i-1+W/2} is a vector)?
10. Line 279, standard deviation and variance are highly dependent (providing repeated information), is there any specified reason for including both in the key property feature s^i?
11. Lines 291-293, why do the vectors s^j, d^j, and g^j in IP vector have the same dimension and how do these vectors construct a matrix? It seems that s^j is a 6x1 vector, d^j is a (n-W-1)x1 vector, and g^j is (n-W-2)x1.
12. Line 340, what is zero-shot pattern? Further explanation or reference is needed here.
13. Line 356, please explain what are 'in-distribution' and 'out-of-distribution' settings. Further explanations are required to demonstrate the difference between these two experiment settings.
14. Line 465, the authors consider the samples from Arxiv and Yelp as natural language datasets. Is it possible the samples from Yelp consist of some of the GPT-generated texts or Arxiv consists of some of the human-written and GPT-moderated samples?
15. The experiments show that the proposed model can perform well in distinguishing two situations -- human-written and LLM-generated cases. I am curious about the generalization on the marginal cases, such as the text is firstly human-written but later modified (rewritten) or translated by LLM. Will these be considered as human-written or LLM-generated?

---

> ### Author Response · Authors · 2024-11-27
> **Response to Reviewer bf21 (Part 1)**
>
> Thanks for your insightful review and suggestions. Here are our detailed point-by-point feedbacks for your questions:
>
> > W1: Some terms, concepts, and figure captions need definitions and explanations for readers to better understand.
>
> > W2: Mathematical notations and derivation need improvement.
>
> Thank you for your suggestions. We have revised our manuscript to enhance its clarity and readability.
>
> ---
>
> > W3: The experimental setup requires enhancement, and further validation is necessary to evaluate its generalizability.
>
> We have added a new Appendix B to provide more details about our experimental setup, particularly focusing on the construction of our datasets, including details such as the number of samples in each dataset, the specific generation prompts we used, and examples of both human-written and AI-generated texts. We also present how we craft the paraphrased datasets, which are used to evaluate the generalizability of Profiler, shown in Figure 4 with the out-of-distribution (OOD) results. The experimental settings in our paper are consistent with those employed in prior studies [1, 2, 3], ensuring comparability and alignment with established methodologies.
>
> ---
>
> > Q1: In lines 69-70, references are needed for independent and correlated features. 'output logits for each token' needs further explanation. Are they feature vectors learned from something? Why are they independent?
>
> To further illustrate the intuition and definitions of the independent and correlated features used in our paper, we have modified the Motivation (Section 3) in our main text, where we added a new motivation example in Figure 2 and moved the original Figure 2 to Figure 6 in Appendix A. This new Figure 2 illustrates the intuition behind our method by comparing text patterns generated by GPT-4-Turbo and Claude-3-Sonnet. As a standard practice when generating texts using LLMs, a prompt is provided to the model. In this example, both GPT-4-Turbo and Claude-3-Sonnet are given the same prompt, "When a three-dimensional object moves relative to an observer, a change occurs on the observer's". Each model then generates new tokens following its intrinsic pattern, i.e., the texts in green and orange, respectively. During the detection phase, a small surrogate model (e.g., GPT-2 in this example) is used to extract features of the generated texts by inferring them token-by-token, and Profiler analyzes the surrogate model’s output logits of those tokens and their cross-entropy losses. The figure shows that given the original prompt (in gray) and part of the generated text (i.e., “perception of” for GPT and “ret inal” for Claude), how Profiler engineers the features. The first feature (i.e., the bar charts in the first column) is the output logits of context. For example, the top-left bar chart shows the output logits of tokens “of”, “the”, and “object”, given the input inside the green dashed box. Ideally, we hope this feature denotes the likelihoods that the model stutters and repeats the previous word “of”, correctly predicts the expected word “the”, and skips a word and fast-forwards to “object”. In contrast, existing techniques only use the logits value of “the”. Observe from the two bar charts in the left column that the two features appear similar, meaning that the probabilities follow a similar pattern. To zoom in, Profiler computes the cross-entropy losses between the current output logits (e.g., the logits for “the”) and the one-hot encodings of the context (e.g., encodings of “of”, “the”, and “object”, respectively), yielding the charts in the second column. Intuitively, this feature makes the probabilities of stuttering, saying-the-right-word, and skipping more prominent by using the ground-truth tokens as a strong reference. Observe that differences start to emerge. **These features are calculated independently for each token and hence called independent features**. In the last column, we further enhance the distinguishability by subtracting neighboring cross-entropy losses. **These features are called correlated features, as they denote relationships between different tokens in the context**.
>
> Though the features in this new motivation example are not identical to those used in Profiler, they help clarify the distinction between independent and correlated features in our method. By leveraging these complementary feature types, Profiler achieves robust and accurate text origin detection.

---

> ### Author Response · Authors · 2024-11-27
> **Response to Reviewer bf21 (Part 2)**
>
> > Q2: In lines 184-186, do the PROFILER features only have 2 dimensions? If not, how are these two dimensions selected?
>
> The features of Profiler are multi-dimensional, as detailed in Section 4.4. Specifically, for one surrogate model, Profiler generates features with 3*6*W+C(W,2) dimensions, where W is the context window size and C(W,2) is calculated as W(W-1)/2. To visualize the effectiveness of Profiler, as stated in Line 183, we employ t-SNE to reduce the dimensionality of these features and select the two most representative ones for visualization.
>
> ---
>
> > Q3: In Figure 2, most of the features have an oval shape. This makes sense since projecting them onto the PROFILER feature axis 1/2 gives you Gaussian distributions. Is there an explanation for why do the GPT-3.5 Turbo features (green) not follow a 2D Gaussian distribution and why does it look very different from GPT-4 Turbo (I do not expect the shapes to be very different from GPTs)?
>
> Thank you for your question. To clarify, the green dots in Figure 2 represent samples generated by GPT-4-Turbo, while the blue dots represent samples generated by GPT-3.5-Turbo. One potential reason why the GPT-4-Turbo samples deviate from an oval shape is the limited amount of data. In Figure 2, we visualize points from the Essay dataset, which contains at most 2,000 samples per model. This limited sample size may not fully capture the complete distribution of an LLM’s generation.
>
> This observation is further supported by the distribution of Binoculars scores in Figure 1, where GPT-4-Turbo's score distribution is the least standard and most asymmetric, corresponding to its distinct feature distribution in Profiler. Similarly, sample dots for other models (e.g., the red Claude-3-Sonnet dots and purple Gemini-1.0-Pro dots) also deviate from standard oval shapes. Such deviations are expected, as real-world sample distributions often differ from ideal distributions when data is limited.
>
> The noticeable distribution differences between GPT-3.5-Turbo and GPT-4-Turbo are further supported by the results in Table 1 and Table 2, where most supervised-trained baselines effectively distinguish between samples generated by these two models. A plausible explanation for this is that GPT-3.5-Turbo and GPT-4-Turbo belong to different generations, likely involving differences in model architectures, training procedures, and other factors, which result in significant variations in their outputs despite being developed by the same organization.
>
> ---
>
> > Q4: It seems PROFILER feature 1 provides most of the separable information, and the feature 2 ranges of different data samples are highly overlapped. Is it possible to separate the texts with only one feature?
>
> Thank you for the question. We want to clarify that the features used by Profiler are automatically selected via t-SNE. The complete features used in Profiler are multidimensional, with 3*6*W+C(W,2) dimensions for each surrogate model. In Figure 2, we visualize only the two most representative features for clarity and readability. Although the features have some overlap, feature 2 and other features that are not shown are complementary to feature 1. We conducted an experiment during rebuttal. Using only feature 1 causes performance degradation of 0.09 in the average detection AUC score (from 0.86 to 0.77), highlighting the importance of leveraging multiple features for robust and accurate detection.
>
> ---
>
> > Q5: In Equation 1, what does the black dot represent? Functions with black dots represent a family of functions and are usually used for caption explanations, but not used for formally defining a variable.
>
> As stated in Section 4.2, the black dot represents the probability distribution of the output logits over the vocabulary list V at each position i. To improve clarity, we have updated the notation from a black dot to the commonly used symbol Y_{i} to represent this distribution. We hope this modification addresses your concern effectively.
>
> ---
>
> > Q6: Lines 235-240, what is the definition of the input token sequence X? X=x_{1:n} or X=x_{1:i}? If X=x_{1:i}, X needs a subscript index i (e.g., X_i), since X depends on I.
>
> Thanks for your suggestions. We have added a subscript index i to X.
>
> ---
>
> > Q7: Line 256, what is the definition of C? Is C a set or a vector?
>
> The symbol C was used as a notation to differentiate our context loss L^C  from L, which is typically used to represent training loss in the machine learning field. However, we understand that the use of C may have caused confusion. To address this, we have removed it in our revised manuscript for improved clarity.

---

> ### Author Response · Authors · 2024-11-27
> **Response to Reviewer bf21 (Part 3)**
>
> > Q8: Lines 259-260, why does an even W guarantee the context matrix L^C to be symmetric? It seems that the dimension of L^C also depends on n.
>
> As stated in Line 259, the context losses have a shape of W by (n-W), where (n-W) represents the length of each loss subsequence, and W corresponds to the context window size. At each position of the output logits sequence, W context loss values are calculated. Consequently, if W is an even number, the context loss L^C will exhibit symmetry at each output logits position.
>
> ---
>
> > Q9: Equation 2 looks confusing. Considering that tilde $P_k$ is a ||V||x1 vector, the dot in Equation 2 is an element-wise product (if $o^v_{i-1+W/2}$ is a scalar) or a vector inner product (if $o^v_{i-1+W/2}$ is a vector)?
>
> Both tilde $P^v_{i-1+j}$ and $o^v_{i-1+w/2}$ are scalars in equation (2), since both of them have a superscript “v”.
>
> ---
>
> > Q10: Line 279, standard deviation and variance are highly dependent (providing repeated information), is there any specified reason for including both in the key property feature $s^i$?
>
> We agree with your point that standard deviation (std) and variance are highly dependent, though they are not exactly the same. Based on our tests, including both features, while seemingly redundant, can slightly improve the overall performance of Profiler. As shown in the table below, we evaluated different settings on the Yelp dataset under the in-distribution setting. The results demonstrate that using both std and variance achieves the highest AUC in most cases, outperforming configurations that use only one of the features.
> |       Setting       |    Human   | GPT-3.5 Turbo | GPT-4 Turbo | Claude-3 Sonnet | Claude-3 Opus | Gemini 1.0-Pro | Average AUC |
> |:-------------------:|:----------:|:-------------:|:-----------:|:---------------:|:-------------:|:--------------:|:-----------:|
> | w/ both var and std | **0.9839** |   **0.8563**  |  **0.8595** |    **0.8513**   |     0.8758    |     0.8471     |  **0.8790** |
> |     Only w/ std     |   0.9834   |     0.8539    |    0.8577   |      0.8507     |   **0.8782**  |     0.8480     |    0.8786   |
> |     Only w/ var     |   0.9835   |     0.8539    |    0.8577   |      0.8508     |   **0.8782**  |   **0.8481**   |    0.8787   |
>
> ---
>
> > Q11: Lines 291-293, why do the vectors $s^j$, $d^j$, and $g^j$ in IP vector have the same dimension and how do these vectors construct a matrix? It seems that $s^j$ is a 6x1 vector, $d^j$ is a (n-W-1)x1 vector, and $g^j$ is (n-W-2)x1.
>
> All the $s^j$, $d^j$, and $g^j$ vectors are 1-D vectors. When crafting the independent patterns (IP), we concatenate all these 1-D feature vectors to form a longer 1-D vector. Specifically, each $s^j$, $d^j$, and $g^j$ is with 6 dimensions, hence the concatenated IP is with 3*6*W dimensions. We noticed that the expression in Line 291 might cause confusion, so we have fixed the expression here in our revised manuscript.
>
> ---
>
> > Q12: Line 340, what is zero-shot pattern? Further explanation or reference is needed here.
>
> The zero-shot pattern represents the detection pattern employed by zero-shot detection methods. Specifically, these methods typically assign a probability score to a given text, estimating how likely it is to be generated by a specific source LLMs. This is achieved using statistical metrics on the output logits from surrogate models, typically without requiring any fine-tuning.
>
> For RADAR and the OpenAI Detector, while their original methodologies involve fine-tuning a small language model to distinguish between human-written and AI-generated texts across various source LLMs, we utilized their officially released detection models. Instead of performing further fine-tuning, we directly fed input texts into their pre-trained detector models to obtain probability scores. This approach aligns with a zero-shot detection methodology in practice.
>
> We have clarified these details in our revised manuscript to ensure greater transparency and understanding.
>
> ---
>
> > Q13: Line 356, please explain what are 'in-distribution' and 'out-of-distribution' settings. Further explanations are required to demonstrate the difference between these two experiment settings.
>
> Thanks for your suggestion. We have added these details in Section 5.2 in our revised manuscript. Briefly speaking, the in-distribution setting represents that the distribution of the training set and test set are the same (e.g., we train and test the detector both on GPT-3.5-Turbo-generated data), while the out-of-distribution setting represents that the distribution of the training set and test set are distinct (e.g, the detector is trained on the normal dataset while tested on the paraphrased dataset).

---

> ### Author Response · Authors · 2024-11-27
> **Response to Reviewer bf21 (Part 4)**
>
> > Q14: Line 465, the authors consider the samples from Arxiv and Yelp as natural language datasets. Is it possible the samples from Yelp consist of some of the GPT-generated texts or Arxiv consists of some of the human-written and GPT-moderated samples?
>
> The human-written data in both the Arxiv and Yelp datasets are sourced from existing studies [1], where the data were collected from papers or posts created before commercial LLMs became publicly accessible. As a result, the human-written samples in these datasets are not GPT-moderated.
>
> ---
>
> > Q15: The experiments show that the proposed model can perform well in distinguishing two situations -- human-written and LLM-generated cases. I am curious about the generalization on the marginal cases, such as the text is firstly human-written but later modified (rewritten) or translated by LLM. Will these be considered as human-written or LLM-generated?
>
> Thank you for raising this interesting issue. Classifying AI-modified human samples is indeed an unresolved question. Without an official definition from governments or international organizations, it is challenging to address this task in academia, as such a definition is closely tied to real-world applications and ethical standards.
>
> We believe that as research in AI-generated text detection and text origin detection continues to gain influence, a clearer definition of these mixed samples will emerge, enabling future work to tackle this task effectively. However, in this paper, as well as in most existing studies, we do not consider this mixed case.
>
> ---
>
> **References**
>
> 1. Mao, Chengzhi, et al. "Raidar: geneRative AI Detection viA Rewriting." International Conference on Learning Representations (ICLR). 2024.
> 2. Verma, Vivek, et al. "Ghostbuster: Detecting text ghostwritten by large language models." Conference of the North American Chapter of the Association for Computational Linguistics: Human Language Technologies (NAACL). 2024.
> 3. Hu, Xiaomengc, Pin-Yu Chen, and Tsung-Yi Ho. "RADAR: Robust AI-text detection via adversarial learning." International Conference on Neural Information Processing Systems (NeurIPS). 2023.

---

> ### Author Response · Authors · 2024-12-02
> **Kind Reminder from Authors**
>
> Dear Reviewer bf21,
>
> We sincerely appreciate your valuable suggestions, which have significantly enhanced the quality of our manuscript. In response to your feedback, we have made every effort to address your concerns regarding the clarity of our design. Specifically, we have revised Section 4 to provide clearer mathematical expressions and explanations, and modified Section 3 to better present our motivation.
>
> We would be very grateful for any further feedback you may have on the revised version and our responses. If there are any aspects that remain unclear, we are more than willing to provide additional clarification.
>
> If our responses have adequately addressed your concerns, we kindly ask you to reconsider the score.
>
> Thank you once again for your time and thoughtful review. We look forward to your response.
>
> Best regards,
>
> The Authors

---

### Official Review · Reviewer_tEky · 2024-11-03

**Soundness:** 2
**Presentation:** 2
**Contribution:** 2
**Rating:** 3
**Confidence:** 4

**Summary:**

This work focused on different AI-generated text origen detection. Compared to other baselines, this work proposed to capture the context-aware patterns between the generated output logits and its adjacent input contexts. By collecting such contextual information across different close-source commercial LLMs, such as GPT-4-Turbo, Claude3 Sonnet and Gemini-1.0 Pro. The proposed method Profiler outperforms several baselines across 6 different datasets.

**Strengths:**

- This work designed a contextual loss between the output logits and its adjacent input tokens, and then use this pattern to further capture the independent and correlated patterns to train a classifier.
- This work evaluated their methods across different baselines and datasets.

**Weaknesses:**

- This work lacks lots of details about how to construct the AI-generated texts from GPT-3.5-Turbo, GPT-4-Turbo, Claude-3-Sonnet, Claude-3-Opus, and Gemini-1.0-Pro, for example, how many data samples are generated for each dataset, how different is each generated sample compared to original dataset samples, and what kinds of prompts are used to instruct those five close-source LLMs?
- It lacks reasonable explanations as to why the cross-entropy loss between the output logits with its adjacent input tokens can capture the difference between different LLMs' generated texts. In addition, there is no more analysis regarding this contextual loss in the experimental results section, and how to make the correlation between the entropy loss and different identified LLMs' generated texts.

**Questions:**

- This work chose the surrogate model to detect different AI-generated texts. In line 55, the authors also mentioned that a surrogate model is an LLM with comparable capabilities. This work uses LLaMA2-7B, LLaMA2-13B, LLaMA3-8B, Mistral-7B, Gemma-2B, Gemma-7B as surrogate models to detect close-source LLMs, such as GPT-3.5-Turbo, GPT-4-Turbo, Claude3-Sonnet, Claude-3-Opus and Gemini-1-Pro. It is interesting whether those surrogate models have comparable capabilities to detect those larger close-source LLMs.
- In line 247, the argument about the potential overlapping training data needs further explanation as we actually do not know what kinds of training data are used for those closed-source LLMs.
- As mentioned in the weakness section, it is unclear why the cross-entropy loss works to detect different LLMs' generated texts. What does the cross-entropy loss represent if the loss is high or low?
- The AI-generated texts lack lots of collection and construction details as mentioned in the weakness section. It is the same for the paraphrased versions of the six datasets. If we do not know how those datasets are constructed, we won't understand why the proposed Profiler method can even achieve close 100% AUC on some datasets for some LLMs, such as Essay dataset for GPT-4 Turbo.
- In line 429, Profiler and other baselines are trained on the original datasets and test them on the paraphrased version of the same datasets. As the used surrogate models are LLaMA2-7B, LLaMA2-13B, LLaMA3-8B, Mistral-7B, Gemma-2B, Gemma-7B, how do authors make sure that those surrogate models never see those datasets before during their pretraining. In addition, do authors train profiler using fine-tuning or other methods? It lacks many details.

---

> ### Author Response · Authors · 2024-11-27
> **Response to Reviewer tEky (Part 1)**
>
> Thanks for your insightful review. Here are our detailed point-by-point feedbacks for your questions:
>
> > W1: This work lacks lots of details about how to construct the AI-generated texts from GPT-3.5-Turbo, GPT-4-Turbo, Claude-3-Sonnet, Claude-3-Opus, and Gemini-1.0-Pro, for example, how many data samples are generated for each dataset, how different is each generated sample compared to original dataset samples, and what kinds of prompts are used to instruct those five close-source LLMs?
>
> Thank you for bringing up this issue. The human-written texts we used were directly sourced from existing papers or open-source datasets, such as Arxiv [1], Yelp [1], Creative [2], Essay [2], HumanEval [3], and GCJ [4, 5]. To generate the corresponding texts using the latest commercial LLMs, we strictly follow the prompts used in existing papers [1, 2, 6], for both normal datasets and paraphrased datasets.
>
> To provide further clarification on how we crafted our datasets, we have added a new Appendix B. This section includes details such as the number of samples in each dataset, the specific generation prompts we used, and examples of both human-written and AI-generated texts. We hope this additional information could address your concerns regarding our dataset construction.
>
> ---
>
> > W2: It lacks reasonable explanations as to why the cross-entropy loss between the output logits with its adjacent input tokens can capture the difference between different LLMs' generated texts. In addition, there is no more analysis regarding this contextual loss in the experimental results section, and how to make the correlation between the entropy loss and different identified LLMs' generated texts.
>
> We have modified the Motivation (Section 3) in our main text, where we added a new motivation example in Figure 2 and we moved the original Figure 2 to Figure 6 in Appendix A. This new Figure 2 illustrates the intuition behind our method by comparing text patterns generated by GPT-4-Turbo and Claude-3-Sonnet. As a standard practice when generating texts using LLMs, a prompt is provided to the model. In this example, both GPT-4-Turbo and Claude-3-Sonnet are given the same prompt, "When a three-dimensional object moves relative to an observer, a change occurs on the observer's". Each model then generates new tokens following its intrinsic pattern, namely, the texts in green and orange, respectively. During the detection phase, a small surrogate model (e.g., GPT-2 in this example) is used to extract features of the generated texts by inferring them token-by-token and analyzing the surrogate model’s output logits of those tokens and their cross-entropy losses. The figure shows that given the original prompt (in gray) and part of the generated text (i.e., “perception of” for GPT and “ret inal” for Claude), how Profiler engineers the features. The first feature is the output logits of context. For example, the top-left bar chart shows the output logits of tokens “of”, “the”, and “object”, given the input inside the green dashed box. Ideally, we hope this feature denotes the likelihoods that the model stutters and repeats the previous word “of”, correctly predicts the expected word “the”, and skips a word and fast-forwards to “object”. In contrast, existing techniques only use the logits value of “the”. Observe from the two bar charts in the left column that the two features appear similar, meaning that the probabilities follow a similar pattern. To zoom in, Profiler computes the cross-entropy losses between the current output logits (e.g., the logits for “the”) and the one-hot encodings of the context (e.g., encodings of “of”, “the”, and “object”, respectively), yielding the charts in the second column. Intuitively, this feature makes the probabilities of stuttering, saying-the-right-word, and skipping more prominent by using the ground-truth tokens as a strong reference. Observe that differences start to emerge. In the last column, we further enhance the distinguishability by subtracting neighboring cross-entropy losses.
>
> ---
>
> > Q1: This work chose the surrogate model to detect different AI-generated texts. In line 55, the authors also mentioned that a surrogate model is an LLM with comparable capabilities. This work uses LLaMA2-7B, LLaMA2-13B, LLaMA3-8B, Mistral-7B, Gemma-2B, Gemma-7B as surrogate models to detect close-source LLMs, such as GPT-3.5-Turbo, GPT-4-Turbo, Claude3-Sonnet, Claude-3-Opus and Gemini-1-Pro. It is interesting whether those surrogate models have comparable capabilities to detect those larger close-source LLMs.
>
> Our description is misleading. We change it to “(i.e., an LLM of a relatively small scale)”. As shown by our study, these models are sufficiently capable and can effectively capture and differentiate the subtle characteristics of human-written and AI-generated texts.

---

> ### Author Response · Authors · 2024-11-27
> **Response to Reviewer tEky (Part 2)**
>
> > Q2: In line 247, the argument about the potential overlapping training data needs further explanation as we actually do not know what kinds of training data are used for those closed-source LLMs.
>
> We acknowledge that it is challenging to clearly identify the overlap between the data used for training different LLMs, particularly for commercial language models where detailed information is not publicly available. However, based on experimental findings from existing studies on extracting pre-training data from production-level LLMs [7, 8] and the official technical reports of open-source LLMs such as LLaMA [9], it is generally believed that these models at least share part of their training data from popular sources (e.g., Wikipedia, GitHub). This potential overlap in pre-training data could partly explain the effectiveness of surrogate-model-based detection methods, as the surrogate models may inherently encode knowledge from similar data distributions. We will add the corresponding references to our main text.
>
> ---
>
> > Q3: As mentioned in the weakness section, it is unclear why the cross-entropy loss works to detect different LLMs' generated texts. What does the cross-entropy loss represent if the loss is high or low?
>
> Please see our response to W2.
>
> ---
>
> > Q4: The AI-generated texts lack lots of collection and construction details as mentioned in the weakness section. It is the same for the paraphrased versions of the six datasets. If we do not know how those datasets are constructed, we won't understand why the proposed Profiler method can even achieve close 100% AUC on some datasets for some LLMs, such as Essay dataset for GPT-4 Turbo.
>
> Thank you for highlighting this issue. We have addressed it in our response to W1 and included a new Appendix B detailing our dataset construction process. The new Appendix B includes details such as the number of samples in each dataset, the specific generation prompts we used, and examples of both human-written and AI-generated texts. We hope this additional information could address your concerns regarding our dataset construction.
>
> The superior performance of our Profiler on the Essay dataset can likely be attributed to its longer text length. As demonstrated in prior studies [1, 2, 3], longer texts typically offer richer contextual information and more distinctive patterns, making them easier to detect compared to shorter texts, such as those in the Arxiv or Yelp datasets. Our experimental results align with this observation. Additionally, other baseline methods also show improved performance with longer texts, such as on the Essay dataset, though they consistently achieve lower AUC scores than Profiler.
>
> ---
>
> > Q5: In line 429, Profiler and other baselines are trained on the original datasets and test them on the paraphrased version of the same datasets. As the used surrogate models are LLaMA2-7B, LLaMA2-13B, LLaMA3-8B, Mistral-7B, Gemma-2B, Gemma-7B, how do authors make sure that those surrogate models never see those datasets before during their pretraining. In addition, do authors train profiler using fine-tuning or other methods? It lacks many details.
>
> For the first part of the question, the human-written texts in the six datasets are well-established and widely studied in existing research [1, 2, 6]. According to both prior studies and the technical reports of the open-source models used in Profiler, these datasets are not typically included in LLM pre-training. Moreover, even if portions of the human-written data were used during the pre-training of the surrogate models, the extracted features from the surrogate model’s output logits would likely be more aligned with those of AI-generated text, making detection harder rather than easier. Importantly, the supervised-training based  baselines in our paper, such as Sniffer and SeqXGPT, also utilize the same surrogate models as Profiler and have a very similar setup, yet they still perform worse, further demonstrating the effectiveness of our approach.
>
> For the second part of the question, we do not fine-tune the surrogate models due to the significant computational costs and potential limitations in real-world scenarios where fine-tuned, dedicated LLMs may not be available. Instead, Profiler trains a lightweight classifier (e.g., random forest in our experiments) to learn and distinguish the inference patterns of texts from different sources, as extracted by the surrogate models, shown in the last paragraph in Section 4.4. This design ensures computational efficiency and the broader applicability of our method without sacrificing detection performance.

---

> ### Author Response · Authors · 2024-11-27
> **Response to Reviewer tEky (References)**
>
> **References**
>
> ---
>
> 1. Mao, Chengzhi, et al. "Raidar: geneRative AI Detection viA Rewriting." International Conference on Learning Representations (ICLR). 2024.
> 2. Verma, Vivek, et al. "Ghostbuster: Detecting text ghostwritten by large language models." Conference of the North American Chapter of the Association for Computational Linguistics: Human Language Technologies (NAACL). 2024.
> 3. Chen, Mark, et al. "Evaluating large language models trained on code." arXiv preprint arXiv:2107.03374 (2021).
> 4. Petrik, Juraj, and Daniela Chuda. "The effect of time drift in source code authorship attribution: Time drifting in source code-stylochronometry." International Conference on Computer Systems and Technologies (CompSysTech). 2021.
> 5. Google. "Google code jam, kickstart and hash code competitions". 2008-2020.
> 6. Hu, Xiaomengc, Pin-Yu Chen, and Tsung-Yi Ho. "RADAR: Robust AI-text detection via adversarial learning." International Conference on Neural Information Processing Systems (NeurIPS). 2023.
> 7. Carlini, Nicholas, et al. "Extracting training data from large language models." USENIX Security Symposium. 2021.
> 8. Nasr, Milad, et al. "Scalable extraction of training data from (production) language models." arXiv preprint arXiv:2311.17035 (2023).
> 9. Touvron, Hugo, et al. "Llama: Open and efficient foundation language models." arXiv preprint arXiv:2302.13971 (2023).

---

> ### Author Response · Authors · 2024-12-02
> **Kind Reminder from Authors**
>
> Dear Reviewer tEky,
>
> We sincerely appreciate your valuable suggestions, which have significantly enhanced the quality of our manuscript. In response to your feedback, we have made our best effort to address your concerns regarding dataset construction and the motivation behind our method. Specifically, we have added a new Appendix B with detailed information on dataset construction and revised Section 3.
>
> We would be very grateful for any further feedback you may have on the revised version and our responses. If there are any aspects that remain unclear, we are more than willing to provide additional clarification.
>
> If our responses have adequately addressed your concerns, we kindly ask you to reconsider the score.
>
> Thank you once again for your time and thoughtful review. We look forward to your response.
>
> Best regards,
>
> The Authors

---

### Official Review · Reviewer_hAaV · 2024-11-04

**Soundness:** 3
**Presentation:** 3
**Contribution:** 3
**Rating:** 6
**Confidence:** 4

**Summary:**

In this paper, the authors present an AI-generated text origin detection method (aka Profiler) by extracting distinct context inference patterns through calculating and analyzing novel context losses between the surrogate model’s output logits and the adjacent input context. They demonstrate the effectiveness of Profiler by comparison against multiple baselines on natural language and code datasets.

**Strengths:**

The experiments are thoroughly performed and the comparison with multiple state-of-the-art baselines bring out the novelty and the advancement clearly. They further present the ablation study to demonstrate the effectiveness of the different components in the proposed architecture such as context window size and surrogate model selection.

**Weaknesses:**

It would really help the readers if the authors can provide the intuition behind the design of Profiler (Section 4). The section, though presents the working of the different components, fails to provide the different design choices behind each component. In the current form, it is difficult to intuitively understand why the proposed approach is working effectively.

**Questions:**

Check my comments in Weaknesses

---

> ### Author Response · Authors · 2024-11-27
> **Response to Reviewer hAaV**
>
> Thanks for your appreciation and suggestions. Here are our point-by-point responses:
>
> > W1: It would really help the readers if the authors can provide the intuition behind the design of Profiler (Section 4). The section, though presents the working of the different components, fails to provide the different design choices behind each component. In the current form, it is difficult to intuitively understand why the proposed approach is working effectively.
>
> We have modified the Motivation (Section 3) in our main text, where we added a new motivation example in Figure 2 and we moved the original Figure 2 to Figure 6 in Appendix A. This new Figure 2 illustrates the intuition behind our method by comparing text patterns generated by GPT-4-Turbo and Claude-3-Sonnet. As a standard practice when generating texts using LLMs, a prompt is provided to the model. In this example, both GPT-4-Turbo and Claude-3-Sonnet are given the same prompt, "When a three-dimensional object moves relative to an observer, a change occurs on the observer's". Each model then generates new tokens following its intrinsic pattern, i.e., the texts in green and orange, respectively. During the detection phase, a small surrogate model (e.g., GPT-2 in this example) is used to extract features of the generated texts by inferring them token-by-token, and Profiler analyzes the surrogate model’s output logits of those tokens and their cross-entropy losses. The figure shows that given the original prompt (in gray) and part of the generated text (i.e., “perception of” for GPT and “ret inal” for Claude), how Profiler engineers the features. The first feature (i.e., the bar charts in the first column) is the output logits of context. For example, the top-left bar chart shows the output logits of tokens “of”, “the”, and “object”, given the input inside the green dashed box. Ideally, we hope this feature denotes the likelihoods that the model stutters and repeats the previous word “of”, correctly predicts the expected word “the”, and skips a word and fast-forwards to “object”. In contrast, existing techniques only use the logit value of “the”. Observe from the two bar charts in the left column that the two features appear similar, meaning that the probabilities follow a similar pattern. To zoom in, Profiler computes the cross-entropy losses between the current output logits (e.g., the logits for “the”) and the one-hot encodings of the context (e.g., encodings of “of”, “the”, and “object”, respectively), yielding the charts in the second column. Intuitively, this feature makes the probabilities of stuttering, saying-the-right-word, and skipping more prominent by using the ground-truth tokens as a strong reference. Observe that differences start to emerge. In the last column, we further enhance the distinguishability by subtracting neighboring cross-entropy losses.

---

> ### Author Response · Authors · 2024-12-02
> **Kind Reminder from Authors**
>
> Dear Reviewer hAaV,
>
> We would like to express our sincere appreciation for your valuable suggestions, which have significantly improved the quality of our manuscript. In response to your feedback, we have made our best effort to address your concerns about the intuition behind our design by rewriting the motivation section and including a new motivation example in Figure 2.
>
> We would be grateful for any further feedback you may have on the revised version and our responses. If there are any aspects that remain unclear, we are more than willing to provide additional clarification.
>
> Thank you once again for your time and thoughtful review. We look forward to your response.
>
> Best regards,
>
> The Authors

---

### Author Response · Authors · 2024-11-27
**General Response**

We sincerely thank all the reviewers for your thoughtful and constructive feedback! We are delighted that you found our work to be novel and "rigorously tested," and we appreciate your recognition of our thorough evaluation, high performance, and strong versatility.

To address your concerns, we have provided detailed, point-by-point responses to each review, offering additional evidence to support our proposed method. Furthermore, we have revised both the main text and the appendix to enhance the clarity of the intuition behind our approach, improve the readability of the mathematical equations, and provide a more comprehensive explanation of our dataset construction process.

Below is a summary of the supplementary information included in the rebuttal materials:

1. We refined the Introduction (Section 1) to clarify the definitions of “independent” and “correlated” features, reducing any potential confusion.
2. We revised the Motivation (Section 3) to provide more intuitive explanations behind the design of Profiler.
3. We polished the mathematical notations and symbols in the Design (Section 4) to improve readability and rigor.
4. We added more detailed explanations of the evaluation setup in Section 5.
5. A new Appendix B was included to provide a detailed explanation of the dataset construction process and more concrete examples of the datasets.

All the revised sections are highlighted in blue in the updated manuscript.

We hope our responses address your concerns and look forward to your further feedback. Thank you again for your valuable comments and recognition of our work.

---

### Author Response · Authors · 2024-12-02
**Kind Reminder from Authors**

Dear Reviewers,

Thank you very much for your valuable efforts in reviewing our manuscript. Just a kind reminder that the discussion period is closing soon. If there are any unclear points regarding our manuscript or rebuttal materials, we are more than happy to provide further clarification.

Best regards,

The Authors

---

### Meta-Review · Area_Chair_srHt · 2024-12-20

**Metareview:**

This paper tackles the problem of identifying the origin of AI-generated texts, a challenge exacerbated by the advanced capabilities of LLMs and the similarities in their outputs. Existing detection techniques often fail to reliably determine the specific source model. To address this, the authors introduce PROFILER, a new black-box detection method that identifies a text's origin by examining unique context inference patterns, specifically through the calculation of context losses between a surrogate model’s output logits and the surrounding input contexts. It effectively differentiates texts from various close-source commercial LLMs (e.g., GPT-4-Turbo, Claude 3, Sonnet, Gemini 1.0 Pro) and outperforms baselines across six datasets.

Strength:
- The experiments are thorough, with comparisons against ten state-of-the-art baselines  withover a 25% average increase in AUC scores across evaluations in detecting the origin of AI-generated texts.
- The method is effective across both natural language and code datasets, showcasing adaptability to various content types.

Weakness:
After reviewing the authors' rebuttal, most weaknesses have been addressed to varying degrees, but I believe there are still some significant weakness remain for improvement:

- While the authors clarified technical aspects of PROFILER (e.g., feature dimensionality, t-SNE, and the use of surrogate models), they did not fully provide intuitive explanations for why certain design choices (e.g., context loss) are effective. More analysis or ablation might help with this.

- The authors did not thoroughly explain why cross-entropy loss effectively captures differences between LLM-generated texts. This fundamental aspect of the methodology remains unclear.

- The paper still does not explore scenarios involving mixed human and LLM-generated texts (e.g., human-written texts modified by LLMs), leaving questions about the generalizability of PROFILER's approach.

- While the authors provided a plausible explanation for the lower performance on the Claude family models, their argument relies heavily on assumptions about similarities between the Claude models without offering concrete supporting evidence.

**Additional Comments On Reviewer Discussion:**

The authors acknowledged the challenges of addressing mixed samples and emphasized the need for clearer definitions. However, this gap limits the practical applicability of the work to real-world detection scenarios. While the authors made commendable efforts to address reviewer concerns, these remaining weaknesses suggest that further refinements and analysis are needed for a more comprehensive contribution.

---

### Decision · Program_Chairs · 2025-01-22

Reject